# TYK2 signaling promotes the development of autoreactive CD8+ cytotoxic T lymphocytes and type 1 diabetes

Keiichiro Mine [1,2] ✉, Seiho Nagafuchi [1], Satoru Akazawa [3], Norio Abiru[3,4], Hitoe Mori[1], Hironori Kurisaki[5], Kazuya Shimoda [6], Yasunobu Yoshikai[2], Hirokazu Takahashi [1,7] & Keizo Anzai[1]

Tyrosine kinase 2 (TYK2), a member of the JAK family, has attracted attention as a potential therapeutic target for autoimmune diseases. However, the role of TYK2 in CD8+ T cells and autoimmune type 1 diabetes (T1D) is poorly understood. In this study, we generate *Tyk2* gene knockout non-obese diabetes (NOD) mice and demonstrate that the loss of *Tyk2* inhibits the development of autoreactive CD8+ T-BET+ cytotoxic T lymphocytes (CTLs) by impairing IL-12 signaling in CD8+ T cells and the CD8+ resident dendritic cell-driven cross-priming of CTLs in the pancreatic lymph node (PLN). *Tyk2*-deficient CTLs display reduced cytotoxicity. Increased inflammatory responses in β-cells with aging are dampened by *Tyk2* deficiency. Furthermore, treatment with BMS-986165, a selective TYK2 inhibitor, inhibits the expansion of T-BET+ CTLs, inflammation in β-cells and the onset of autoimmune T1D in NOD mice. Thus, our study reveals the diverse roles of TYK2 in driving the pathogenesis of T1D.

The immune system is strictly regulated to maintain homeostasis by balancing immune tolerance and immunogenicity, and therefore, a break in this balance leads to autoimmune disorders or increased susceptibility to infectious agents[1]. It is thought that a combination of genetic and environmental factors contributes to the pathogenesis of autoimmune diseases. Multifaceted approaches, although challenging, are needed to uncover the pathogenic mechanisms of autoimmune disease and to develop effective preventive/therapeutic measures against multifactorial diseases[2].

Type 1 diabetes (T1D) is caused by the destruction of insulin-producing pancreatic β-cells and requires lifelong insulin therapy. The number of patients with T1D worldwide in 2021 was approximately 8.4 million and is expected to increase rapidly because of changes in environment and lifestyle[3]. T1D is a multifactorial disease related to genetic and environmental factors with a heterogeneous and complex pathogenesis[4]. Genetic studies of T1D revealed risk loci associated with immune responses[5,6]. A candidate susceptibility gene of T1D is tyrosine kinase 2 (TYK2)[5,6], a member of the Janus kinase (JAK) family. TYK2 is expressed ubiquitously and involves in the signal transduction of type 1 interferons (IFNs), interleukin (IL) −23, IL-12, IL-10, and IL-6[7]. When these cytokines bind to cell surface receptors, TYK2 is activated, phosphorylates the intracellular tail of cytokine receptors, and recruits signal transducers and activators of transcriptions (STATs) in concert with other JAKs. The recruited STATs are phosphorylated, dimerized, and translocated to the nucleus to regulate cytokine-specific gene expression and cytokine secretion[8]. These TYK2 responses confer host defense against microorganisms[9–11]. Of note, *TYK2*-deficient patients have impaired type 1 IFNs and IL-12 signal transduction, resulting in mycobacterial and viral infections[12].

[1]Division of Metabolism and Endocrinology, Faculty of Medicine, Saga University, Saga, Japan. [2]Division of Host Defense, Medical Institute of Bioregulation, Kyushu University, Fukuoka, Japan. [3]Department of Endocrinology and Metabolism, Unit of Translational Medicine, Nagasaki University Graduate School of Biomedical Sciences, Nagasaki, Japan. [4]Midori Clinic, Nagasaki, Japan. [5]Department of Medical Science and Technology, Graduate School of Medical Sciences, Kyushu University, Fukuoka, Japan. [6]Division of Hematology, Diabetes, and Endocrinology, Department of Internal Medicine, Faculty of Medicine, University of Miyazaki, Miyazaki, Japan. [7]Liver Center, Saga University Hospital, Saga University, Saga, Japan. ✉e-mail: sv7899@cc.saga-u.ac.jp

Viruses are reported to be potential environmental factors related to T1D[13–15]. Viral infection may contribute to the development of T1D by several mechanisms including direct β-cell destruction, triggering autoimmunity to β-cells, molecular mimicry, and induction of β-cell dedifferentiation[13,14,16]. We previously reported that reduced *Tyk2* expression in β-cells led to impaired antiviral defense, increasing the risk of β-cell-tropic virus-induced diabetes in mice[11]. In addition, we found that a polymorphism in the promoter region of the *TYK2* gene (ClinVar, 440728), which decreased promoter activity, was a risk factor for T1D, particularly in patients with T1D associated with flu-like syndrome and who were autoantibody negative at the onset of disease[17,18]. Although the role of TYK2 in virus-induced β-cell destruction is controversial[11,19], these observations suggest that *TYK2* is involved in the pathogenesis of diabetes induced by virus.

However, TYK2 signaling is associated with the pathogenesis of autoimmune diseases. Indeed, the loss-of-function mutations and inhibition of TYK2 suppressed autoimmune processes in mice and humans[20–23]. The role of TYK2 in psoriasis, which predominantly manifests as skin lesions[24], is well characterized. In a mouse model of imiquimod-induced psoriasis-like dermatitis, IL-17 and IL-22 secreted from γδ T cells and Th17 upon IL-23 stimulation were involved in skin inflammation and keratinocyte activation, which were improved by TYK2 inhibitor treatment[20,25]. In patients with psoriasis, oral treatment with a selective TYK2 inhibitor, BMS-986165 (deucravacitinib), was efficacious[26]. Furthermore, a TYK2 inhibitor ameliorated disease progression in murine models of spondyloarthritis, lupus nephritis, and inflammatory bowel disease[21,22]. These studies demonstrate the involvement of TYK2 in the development of autoimmune disease and its therapeutic potential, although the precise role of TYK2 in autoimmune T1D remains unknown.

In this study, we generated *Tyk2* knockout (KO) NOD mice and revealed the role of TYK2 in driving the pathogenesis of autoimmune T1D. These findings may provide insight into the development of safe and effective prevention strategies for T1D.

## Results

### *Tyk2* deficiency suppresses autoimmune T1D in NOD mice

To determine the role of TYK2 in autoimmune T1D, we generated *Tyk2* gene KO NOD mice by backcrossing *Tyk2*KO.B6 mice onto the NOD background for 10 generations. We analyzed 64 short tandem repeat (STR) loci in *Tyk2*KO.NOD mice and confirmed that all tested loci except for D9Mit83, which is located in chromosome 9 that has *Tyk2* gene, were of NOD origin (Supplementary Fig. 1a). In addition, we confirmed STRs at the insulin-dependent diabetes susceptibility (Idd) loci (Idd1 to Idd15) were of NOD origin (Supplementary Fig. 2b)[27]. Because the incidence of diabetes is more prevalent in female NOD mice[28], this study used female mice. We analyzed the spontaneous incidence of diabetes in mice (+/+, wild type; +/−, heterozygous *Tyk2*KO; −/−, homozygous *Tyk2*KO) in cohoused conditions after weaning to minimize the effect of different gut microbiota composition on diabetes development. As expected, the incidence of diabetes was reduced in *Tyk2*−/− mice (15.3%) compared with their *Tyk2*+/+ (78.5%) and *Tyk2*+/− (90.0%) littermates (Fig. 1a). In addition, the timing of diabetes onset was delayed in *Tyk2*−/− mice (22 weeks of age (22w)) compared with their *Tyk2*+/+ (15w) and *Tyk2*+/− (16w) littermates (Fig. 1a). Comparable diabetes development was observed between *Tyk2*+/+ and *Tyk2*+/− mice. Although TYK2 was reported to be associated with obesity[29], our colony exhibited equivalent weight gain throughout the study (Supplementary Fig. 2c). At 14w, a reduced number of islets with insulitis were observed in *Tyk2*−/− mice compared with age-matched *Tyk2*-sufficient littermates, consistent with the reduction of diabetes incidence in *Tyk2*−/− NOD mice (Fig. 1b and Supplementary Fig. 1d). Serum anti-insulin autoantibodies (IAAs) were comparable among 14w mice irrespective of genotypes (Fig. 1c). We also analyzed

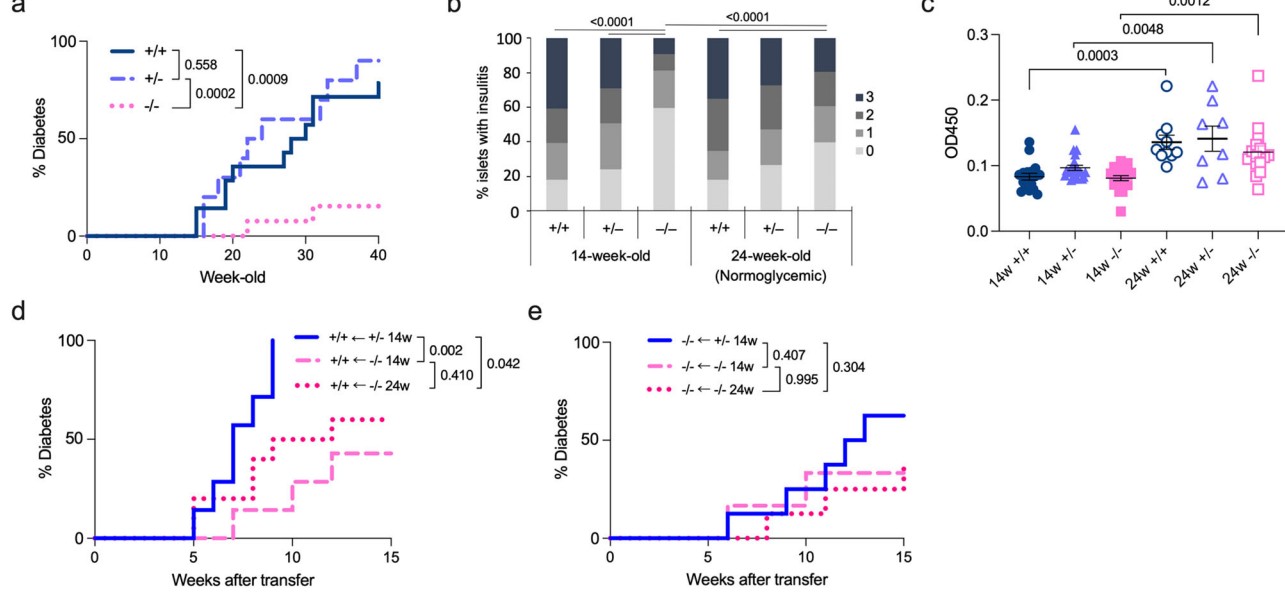

**Fig. 1 | *Tyk2* deficiency suppresses autoimmune diabetes in NOD mice.**
**a** Incidence of spontaneous diabetes in female *Tyk2*+/+ (n = 14), *Tyk2*+/− (n = 10), and *Tyk2*−/− NOD mice (n = 13). Diabetes was defined by a non-fasting blood glucose level exceeding 250 mg/dL. **b** Insulitis scores of normoglycemic mice at 14w (*Tyk2*+/+, n = 8; *Tyk2*+/−, n = 8; *Tyk2*−/−, n = 8) and 24w (*Tyk2*+/+, n = 10; *Tyk2*+/−, n = 8; *Tyk2*−/−, n = 10). Score 0, no insulitis; Score 1, peri-insulitis; Score 2, infiltrative insulitis less than 50% of the islet area; and Score 3, infiltrative insulitis more than 50% of the islet area (Supplementary Fig. 1d). **c** Insulin autoantibody levels in the serum of normoglycemic mice at 14w (*Tyk2*+/+, n = 17; *Tyk2*+/−, n = 22; *Tyk2*−/−, n = 21) and 24w (*Tyk2*+/+, n = 10; *Tyk2*+/−, n = 8; *Tyk2*−/−, n = 16). **d** Incidence of diabetes in female wild type NOD.SCID mice adoptively transferred with splenocytes from normoglycemic female 14w *Tyk2*+/− NOD mice (n = 7), 14w *Tyk2*−/− NOD mice (n = 7), or 24w *Tyk2*−/− NOD mice (n = 10). Splenocytes (1 ×10⁷) were intravenously transferred into 6−8w recipient mice. Diabetes was defined by a non-fasting blood glucose level exceeding 250 mg/dL. **e** Incidence of diabetes in female *Tyk2*−/− NOD.SCID mice adoptively transferred with splenocytes (1 ×10⁷) from normoglycemic female 14w *Tyk2*+/− NOD mice (n = 8), 14w *Tyk2*−/− NOD mice (n = 6), or 24w *Tyk2*−/− NOD mice (n = 8). Data represent the mean ± SEM. *P*-values were calculated using the log-rank test (**a**, **d**, **e**), and one-way ANOVA with Tukey's posttest (**b**, **c**).

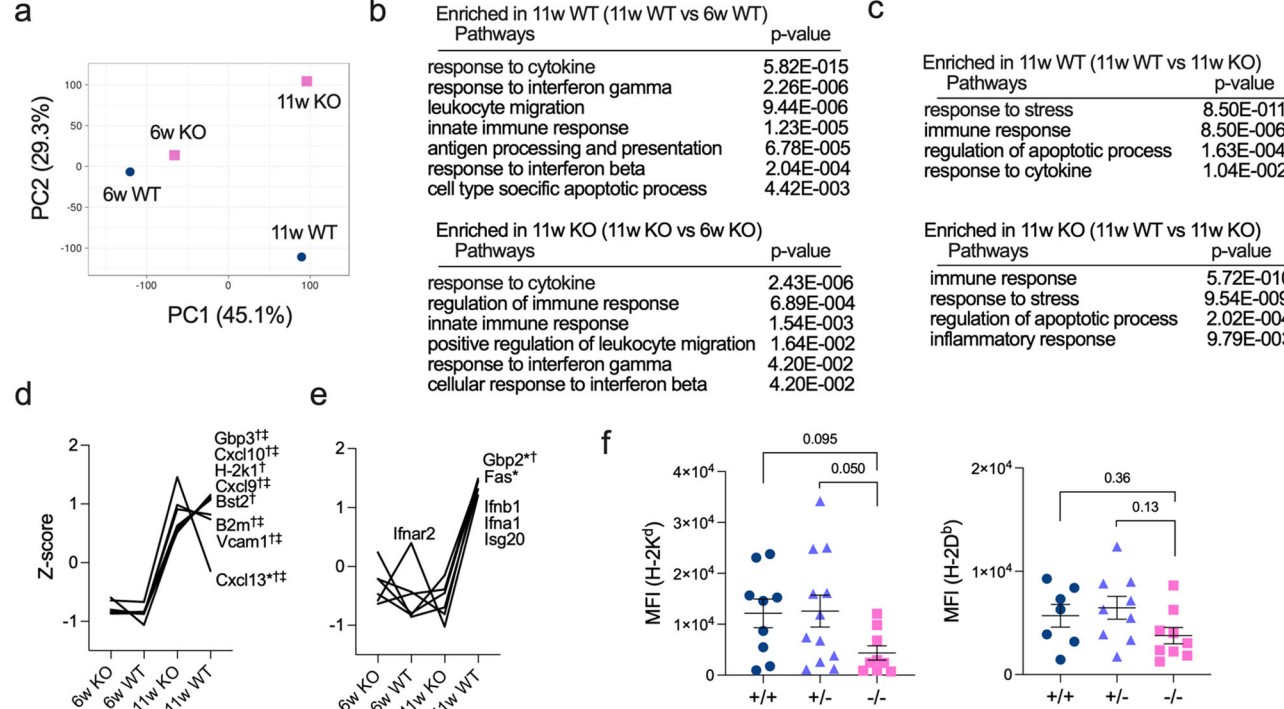

**Fig. 2 | Increased inflammatory signature in β-cells with age is attenuated by *Tyk2* deficiency. a** Principal component analysis (PCA) of transcriptome data of purified β-cells pooled from 6w female *Tyk2*⁺/⁺ (n = 6), 6w female *Tyk2*⁻/⁻ (n = 6), 11w female *Tyk2*⁺/⁺ (n = 8), and 11w female *Tyk2*⁻/⁻ NOD mice (n = 9). β-cells were purified using FluoZin-3-AM and TMRE, and analyzed for gene expression profiles. **b** Gene ontology (GO) biological process enrichment analysis of differentially expressed genes between β-cells from 11w mice and 6w mice (11w *Tyk2*⁺/⁺ vs 6w *Tyk2*⁺/⁺, or 11w *Tyk2*⁻/⁻ vs 6w *Tyk2*⁻/⁻). Selected GO terms are shown. The two-sided adjusted *p*-values were calculated using Benjamini-Yekutieli method (FDR *p* < 0.05) by Gene-Trail. **c** GO biological process enrichment analysis of differentially expressed genes in β-cells from 11w mice (11w *Tyk2*⁺/⁺ vs 11w *Tyk2*⁻/⁻). Selected GO terms are shown. The two-sided adjusted *p*-values were calculated using Benjamini-Yekutieli method (FDR *p* < 0.05) by GeneTrail. **d**, **e** Gene expression profiles in purified β-cells. Data are normalized by Z score. Selected genes are shown. Asterisks (*) indicate differentially expressed genes between 11w wild type vs 11w KO mice. Daggers (†) indicate differentially expressed genes between 6w wild type vs 11w wild type mice. Double daggers (‡) indicate differentially expressed genes between 6w KO vs 11w KO mice. Criteria used for identifying differentially expressed genes were as follows: upregulated genes, Z score >2.0 and ratio <1.5; downregulated genes, Z score <−2.0 and ratio >0.66. **f** Protein expression levels of MHC I (H-2K^d (*Tyk2*⁺/⁺, n = 9; *Tyk2*⁻/⁻, n = 12; *Tyk2*⁻/⁻, n = 9) and H-2D^b (*Tyk2*⁺/⁺, n = 7; *Tyk2*⁺/⁻, n = 9; *Tyk2*⁻/⁻, n = 9)) in β-cells from 11–12w female mice. Data represent the mean ± SEM. *P*-values were calculated using Kruskal-Wallis test with Dunn's posttest (**f**).

---

normoglycemic 24w mice to assess the long-term effects of *Tyk2* deficiency. Increased numbers of inflamed islets and elevated levels of serum IAAs were observed in 24w *Tyk2*⁻/⁻ mice compared with 14w *Tyk2*⁻/⁻ mice (Fig. 1b, c). Because diabetic mice were excluded, levels of insulitis and IAAs were understated in this analysis, especially in 24w *Tyk2*⁺/⁺ and *Tyk2*⁺/⁻ mice. These observations suggest that *Tyk2* deficiency does not completely prevent islet autoimmunity but reduces the progression rate of invasive insulitis leading to T1D onset.

## TYK2 expression in immune cells and the host environment is associated with the development of autoimmune T1D

We next asked whether TYK2 expression was critical for immune cells and/or the host environment (other than immune cells) using an adoptive transfer model. Because *Tyk2*⁺/⁺ and *Tyk2*⁺/⁻ mice had equivalent diabetes incidence (Fig. 1a) and *Tyk2* KO allele had neomycin-resistant cassette[30], we used splenocytes derived from *Tyk2*⁺/⁻ mice as a positive control. The phenotype of the transferred splenocytes, defined by CD44 and CD62L expressions (CD44^lo CD62L⁺, naïve; CD44^hi CD62L⁻, effector memory; CD44^hi CD62L⁺, central memory), were comparable (Supplementary Fig. 1e). NOD.SCID mice, which lack T and B cells, adoptively transferred with *Tyk2*⁺/⁻ splenocytes developed diabetes within nine weeks after transfer, whereas mice receiving *Tyk2*⁻/⁻ splenocytes had reduced diabetes incidence irrespective of the donor age (Fig. 1d). Next, to determine the role of TYK2 in the host environment, we generated *Tyk2*-deficient NOD.SCID mice and conducted the transfer experiments. We revealed that a loss of

*Tyk2* in the host environment led to a reduced incidence of diabetes induced by *Tyk2*⁺/⁻ splenocytes compared with those in wild type host NOD.SCID mice (*P* = 0.003) (Fig. 1e). Thus, these results suggest that TYK2 expression in immune cells and in the host environment contributes to the development of autoimmune T1D.

## Reduced *Fas* expression and type I IFN signatures in β-cells from *Tyk2*-deficient mice

Pancreatic β-cells are a source of islet-associated autoantigens; therefore, we focused on β-cells as an important component of the host environment. We performed transcriptome analysis of β-cells purified by FluoZin-3-AM and tetramethylrhodamine ethyl ester perchlorate (TMRE) from at least six female mice to minimize the effect of individual heterogeneity on T1D development[31,32]. β-cells from 6w *Tyk2*⁺/⁺ mice and 6w *Tyk2*⁻/⁻ mice had similar gene expression profiles, whereas β-cells from 11w mice had different gene expression profiles from those of 6w mice (Fig. 2a). This suggests that TYK2 signaling in β-cells becomes active with age. These differences in gene expression profiles in β-cells with age were confirmed by hierarchical clustering that divided the data into two clusters by age (Supplementary Fig. 1f). Gene ontology (GO) analysis revealed that β-cells from 11w *Tyk2*⁺/⁺ and *Tyk2*⁻/⁻ mice were enriched in immune-related pathways compared with those from 6w mice (Fig. 2b). Comparison of β-cells from 11w *Tyk2*⁺/⁺ and 11w *Tyk2*⁻/⁻ mice showed that both genotypes were enriched in immune-related pathways (Fig. 2c). Indeed, inflammation-related genes including *Vcam1*, *B2m*, and interferon stimulated genes

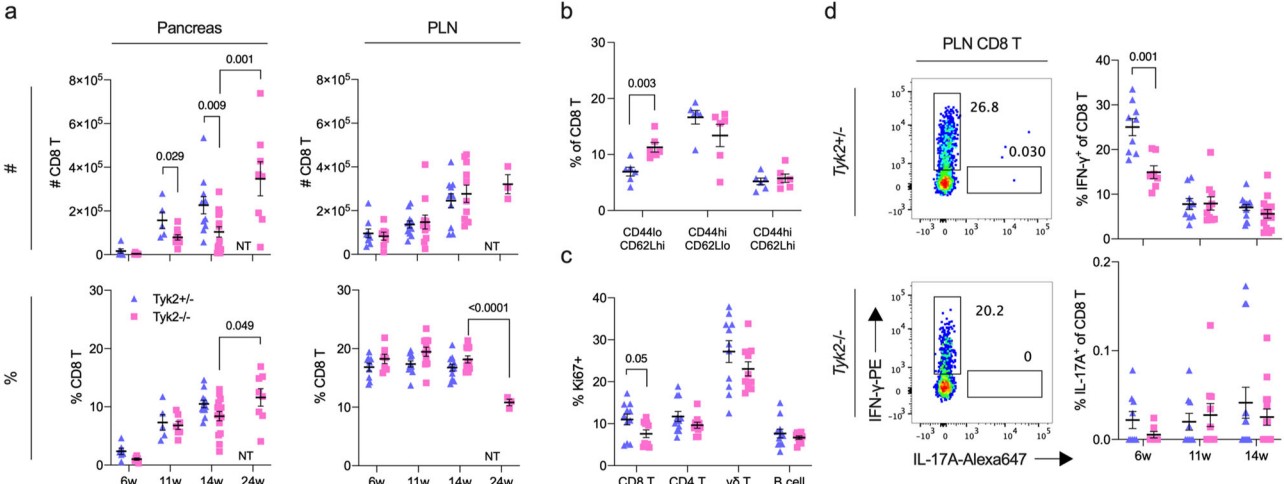

**Fig. 3 | Increased naïve CD8⁺ T cells and reduced IFN-γ-producing CD8⁺ T cells in the PLN of *Tyk2*-deficient NOD mice. a** Cell number (#) or frequency among CD45⁺ cells (%) of CD8⁺ T cells in the PLN and pancreas measured by flow cytometry. Pancreas; 6w (*Tyk2⁺ᐟ⁻*, n = 6; *Tyk2⁻ᐟ⁻*, n = 6), 11w (*Tyk2⁺ᐟ⁻*, n = 5; *Tyk2⁻ᐟ⁻*, n = 9), 14w (*Tyk2⁺ᐟ⁻*, n = 11; *Tyk2⁻ᐟ⁻*, n = 15), and normoglycemic 24w (*Tyk2⁻ᐟ⁻*, n = 8) female NOD mice. PLN; 6w (*Tyk2⁺ᐟ⁻*, n = 9; *Tyk2⁻ᐟ⁻*, n = 7), 11w (*Tyk2⁺ᐟ⁻*, n = 11; *Tyk2⁻ᐟ⁻*, n = 11), 14w (*Tyk2⁺ᐟ⁻*, n = 11; *Tyk2⁻ᐟ⁻*, n = 11), and normoglycemic 24w (*Tyk2⁻ᐟ⁻*, n = 3) female NOD mice. **b** Frequency of naïve (CD44ˡᵒ CD62L⁺), effector memory (CD44ʰⁱ CD62L⁻), and central memory (CD44ʰⁱ CD62L⁺) CD8⁺ T cells in the PLN of 6w female mice (*Tyk2⁺ᐟ⁻*, n = 6; *Tyk2⁻ᐟ⁻*, n = 6). **c** Frequency of Ki67⁺ cells among the indicated immune cells in the PLN of 6w female mice (*Tyk2⁺ᐟ⁻*, n = 11; *Tyk2⁻ᐟ⁻*, n = 10). **d** (Left) Representative flow cytometry plots of IFN-γ and IL-17-producing CD8⁺ T cells upon PMA/iono stimulation in the PLN of 6w female mice. (Right) The graphs indicate the frequency of IFN-γ and IL-17-producing CD8⁺ T cells among CD8⁺ T cells in the PLN at the indicated ages (6w (*Tyk2⁺ᐟ⁻*, n = 9; *Tyk2⁻ᐟ⁻*, n = 7), 11w (*Tyk2⁺ᐟ⁻*, n = 9; *Tyk2⁻ᐟ⁻*, n = 10), and 14w (*Tyk2⁺ᐟ⁻*, n = 14; *Tyk2⁻ᐟ⁻*, n = 14)). NT not tested. Data represent the mean ± SEM. *P*-values were calculated using two-tailed Student's *t* tests.

(ISGs), were highly expressed in β-cells from 11w mice compared with 6w mice irrespective of the *Tyk2* genotypes (Fig. 2d).

However, *Gbp2*, an ISG, was expressed at lower levels in β-cells from 11w *Tyk2⁻ᐟ⁻* mice compared with 11w *Tyk2⁺ᐟ⁺* mice (Fig. 2e). Notably, the expression level of *Fas* (*Cd95*), a cell surface death receptor involved in β-cell death[33], was lower in β-cells from 11w *Tyk2⁻ᐟ⁻* mice compared with 11w *Tyk2⁺ᐟ⁺* mice (Fig. 2e). The expressions of ISGs (*Gbp3*, *Bst2*, and *Isg20*), chemokines (*Cxcl9* and *Cxcl10*), a type I IFN receptor (*Ifnar2*), and type I IFNs (*Ifnb*, *Ifna1*) were slightly lower in β-cells from 11w *Tyk2⁻ᐟ⁻* mice compared with 11w *Tyk2⁺ᐟ⁺* mice, although they did not reach statistical significance (Fig. 2d, e). In contrast, *Cxcl13*, a ligand for CXCR5 involved in B cell chemotaxis, was expressed at higher levels in β-cells from 11w *Tyk2⁻ᐟ⁻* mice compared with 11w *Tyk2⁺ᐟ⁺* mice (Fig. 2d). These results suggest that type I IFN signaling, T cell migration into inflamed islets, and FAS-mediated β-cell death are impaired in β-cells of *Tyk2⁻ᐟ⁻* mice compared with *Tyk2*-sufficient mice.

The increased expression of MHC I in β-cells was reported to be a feature of autoimmune T1D, and TYK2 inhibition prevented IFN-α-induced MHC I expression in human β-cells[19,34]. *H-2k1*, an MHC I molecule, was expressed at lower levels, whereas *B2m*, a component of MHC I, was expressed at higher levels, in β-cells from 11w *Tyk2⁻ᐟ⁻* mice compared with 11w *Tyk2⁺ᐟ⁺* mice (Fig. 2d). We also analyzed the protein expression levels of MHC I in β-cells and revealed that H-2Kᵈ, but not H-2Dᵇ, was slightly decreased in 11–12w *Tyk2⁻ᐟ⁻* mice compared with age-matched littermates (Fig. 2f). Together, these findings suggest that the increased inflammatory state in β-cells with age is attenuated by *Tyk2* deficiency, which may lead to the preservation of β-cells from immune cell attack.

Type I IFN signaling in β-cells was associated with the presentation of autoantigens[19,35]; therefore, we compared the expression levels of T1D-related autoantigens in β-cells. At 11w, gene expression levels of T1D-related autoantigens including insulin (*Ins1* and *Ins2*), GAD65 (*Gad2*), IA-2 (*Ptprn*), ZnT-8 (*Slc30a8*), and IGRP (*G6pc2*) in β-cells were comparable between genotypes (Supplementary Fig. 1g). However, at 6w, *Ins1* and *G6pc2* were expressed at lower levels in β-cells from *Tyk2⁻ᐟ⁻* mice compared with *Tyk2⁺ᐟ⁺* mice (Supplementary Fig. 1g).

## Increased naïve CD8⁺ T cells in the PLN of *Tyk2*-deficient mice

TYK2 was reported to be involved in immune cell functions[30,36], and therefore we assessed immune cell profiles with age. T cell maturation in the thymus was comparable between *Tyk2⁺ᐟ⁻* and *Tyk2⁻ᐟ⁻* mice (Supplementary Fig. 2a). There was no difference in the frequency and number of splenic immune cells (CD4⁺ T, CD8⁺ T, γδ T, and B cells) with some exceptions (Supplementary Fig. 2b). Next, we assessed immune cells in the pancreas (Supplementary Fig. 2c). Although the frequency was equivalent between genotypes, the numbers of CD4⁺ T, CD8⁺ T, γδ T, and B cells in the pancreas were decreased in 11–14w *Tyk2⁻ᐟ⁻* mice compared with *Tyk2⁺ᐟ⁻* mice (Fig. 3a, Supplementary Fig. 2d-f). The phenotypes of CD4⁺ T and CD8⁺ T cells defined by CD44 and CD62L expression levels were also comparable between genotypes (Supplementary Fig. 3a). In addition, the frequency of IFN-γ or IL-17A-producing T cells upon PMA/ionomycin stimulation, which are important effector cells in autoimmune T1D[37], was similar between genotypes (Supplementary Fig. 3b). Thus, there was reduced immune cell migration into the pancreas of *Tyk2⁻ᐟ⁻* mice without cell-type specificity, and these migrated T cells had no defects in IFN-γ and IL-17A production following T cell receptor stimulation.

Pancreatic lymph node (PLN) is a reservoir for islet-autoreactive T cells[38,39]. Next, we analyzed the PLN and found that naïve CD8⁺ T cells were increased in the PLN of 6w *Tyk2⁻ᐟ⁻* mice compared with age-matched *Tyk2⁺ᐟ⁻* mice (Fig. 3b and Supplementary Fig. 3c). Ki67-positive proliferating CD8⁺ T cells were slightly decreased in the PLN of *Tyk2⁻ᐟ⁻* mice compared with age-matched *Tyk2⁺ᐟ⁻* mice (Fig. 3c). In addition, IFN-γ-producing CD8⁺ T cells upon PMA/ionomycin stimulation were decreased in the PLN of 6w *Tyk2⁻ᐟ⁻* mice (Fig. 3d and Supplementary Fig. 3e). These defects were not observed in CD8⁺ T cells from the inguinal LN (iLN) (Supplementary Fig. 3d, f). There was comparable cytokine producing capacity of CD4⁺ T cells and similar frequency of CD4⁺ CD25⁺ FOXP3⁺ Treg in the PLN between the genotypes (Supplementary Fig 3g, h). Although these observations in CD8⁺ T cells were restricted to the initiation phase of autoimmune T1D when the PLN was crucial for the development of diabetes (~10w)[39], these results suggest that the defective proliferation and/or reduced number of CD8⁺ CTLs, which are the primary effector immune cells for β-cell

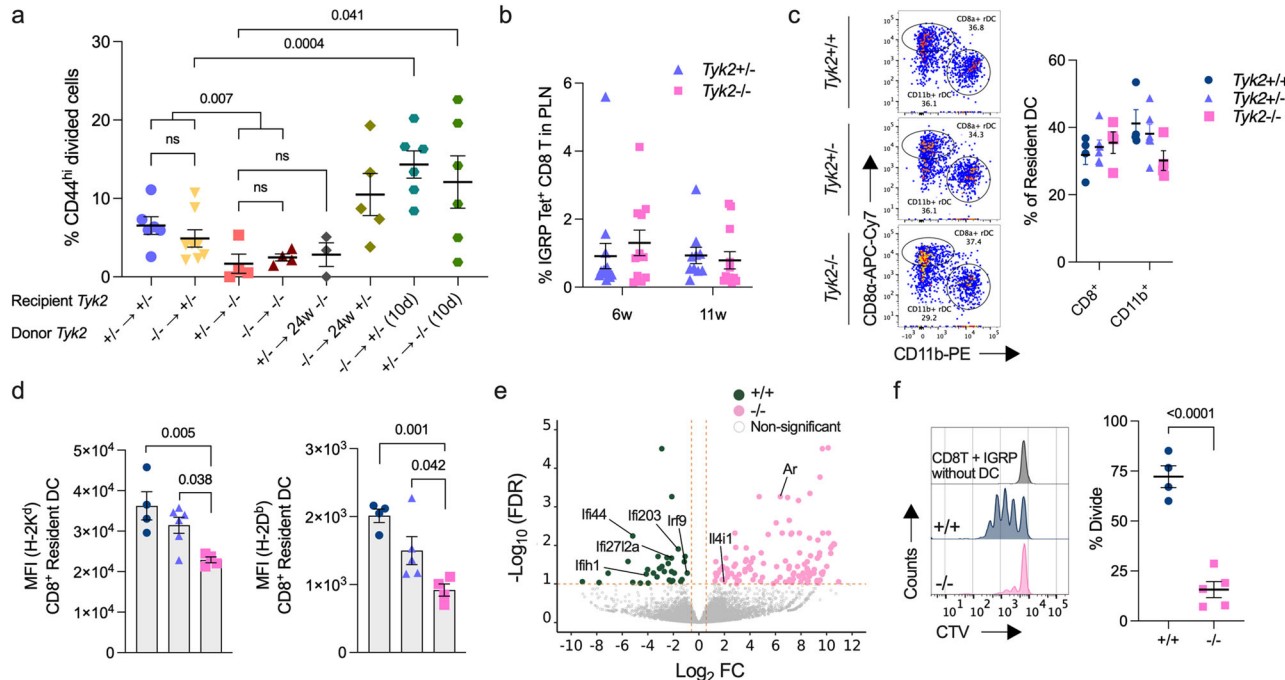

**Fig. 4 | Defective APC function of CD8+ rDC in *Tyk2*-deficient NOD mice.**
**a** Frequency of proliferating CD44hi *Tyk2+/-* or *Tyk2-/-* CTV-labeled 8.3 CD8+ T cells in the PLN of the indicated female recipient mice after 5d or 10d post transfer: 4 ×10^6 labeled naïve 8.3 CD8+ T cells were transferred into 6–8w female recipient mice. "24w" indicates the age of the recipient mice. (*Tyk2+/-* → *Tyk2+/-*, n = 6; *Tyk2-/-* → *Tyk2+/-*, n = 8; *Tyk2+/-* → *Tyk2-/-*, n = 4; *Tyk2-/-* → *Tyk2-/-*, n = 4; *Tyk2+/-* → 24w *Tyk2-/-*, n = 3; *Tyk2-/-* → 24w *Tyk2+/-*, n = 5; *Tyk2-/-* → *Tyk2+/-* 10d, n = 6; *Tyk2+/-* → *Tyk2-/-* 10d, n = 6.) **b** Frequency of IGRP peptide-specific CD8+ T cells in the PLN of female mice (6w (*Tyk2+/-*, n = 14; *Tyk2-/-*, n = 11), 11w (*Tyk2+/-*, n = 10; *Tyk2-/-*, n = 12). **c** (Left) Representative flow cytometry plots of resident DC (rDC) in the PLN of 6w mice. rDCs are gated on MHC IImid CD11chi cells (Supplementary Fig. 5a, b). (Right) The frequencies of CD8+ rDC and CD11b+ rDC among rDC are shown (*Tyk2+/+*, n = 4;

*Tyk2+/-*, n = 6; *Tyk2-/-*, n = 4). **d** Protein expression levels of H-2Kd and H-2Db in CD8+ rDC in the PLN of 6w female mice (*Tyk2+/+*, n = 4; *Tyk2+/-*, n = 6; *Tyk2-/-*, n = 4). **e** Volcano plot of transcriptome data in CD8+ rDC in the PLN of 6w female mice (*Tyk2+/+*, n = 4; *Tyk2-/-*, n = 4). The CD8+ rDC were sorted and RNA-sequencing analysis was performed. Pairwise comparisons of differentially expressed genes were performed using DESeq2 and used to create the plot (FDR < 0.1, logFC > |0.58|). ISGs and genes associated with APC function are highlighted. **f** (Left) Representative histogram and (right) proliferation of CTV labeled naïve 8.3 CD8+ T cells after co-culture with CD8+ rDC and 1 ng/mL of IGRP peptide for 3d (*Tyk2+/+*, n = 4; *Tyk2-/-*, n = 5). Data represent the mean ± SEM. *P*-values were calculated using two-tailed Student's *t* test (**a**, **b**, and **f**) and one-way ANOVA with Dunnett's posttest (**c**, **d**).

destruction[38], in the PLN are responsible for the inhibition of autoimmune T1D development in *Tyk2-/-* NOD mice.

### Reduced proliferation of CD8+ T cells in the PLN of *Tyk2*-deficient mice

Antigen encountered naïve T cells become activated, proliferate, and develop into effector and memory T cells that exhibit the CD44hi phenotype[40]. To investigate the role of TYK2 in the priming of naïve CD8+ T cells, we used islet-specific glucose-6-phosphatase catalytic subunit-related protein (IGRP)-specific CD8+ T cells (8.3 CD8+ T cells) from NY8.3-NOD transgenic mice, which express a T cell receptor (TCR) specific for the islet autoantigen IGRP206-214 peptide[41]. The transfer of *Tyk2+/-* or *Tyk2-/-* naïve 8.3 CD8+ T cells into 6–9w *Tyk2+/-* recipient mice resulted in a similar upregulation of CD44 in the transferred cells, and proliferation in the PLN 5d post transfer, whereas their proliferation was suppressed in the PLN of age-matched *Tyk2-/-* recipient mice (Fig. 4a and Supplementary Fig. 4a, b). These observations suggest that CD8+ T cell-extrinsic mechanisms have a role in the proliferation of islet-autoreactive CTLs in the PLN. Indeed, *Tyk2-/-* 8.3 CD8+ T cells proliferated more in the PLN of *Tyk2+/-* recipient mice 10 days after transfer compared with those 5 days post transfer (Fig. 4a). Ten days after transfer, the number of proliferating 8.3 CD8+ T cells was increased in the PLN of *Tyk2-/-* mice compared with those 5 days after transfer (Fig. 4a), suggesting that the priming of CD8+ T cells was impaired but not abolished in the PLN of *Tyk2-/-* mice. In agreement with a study showing the importance of the PLN for the

proliferation of islet-autoreactive CTLs[39], the limited proliferation of 8.3 CD8+ T cells was observed in the iLN (Supplementary Fig. 4c).

Because IGRP epitopes were reported to appear and spread in the middle-to-late stage of T1D[42], defects in IGRP epitope spreading in *Tyk2-/-* mice might correlate with the reduced proliferation of 8.3 CD8+ T cells in the PLN of *Tyk2-/-* mice. To exclude this possibility, we evaluated IGRP-specific CD8+ T cells (IGRP CD8+ T) using tetramers containing a mimotope of IGRP (NRP-V7). A comparable frequency of IGRP CD8+ T cells was observed in the PLN of both *Tyk2* genotypes (Fig. 4b). Furthermore, the proliferation of 8.3 CD8+ T cells was also suppressed in the PLN of normoglycemic 24w *Tyk2-/-* recipient mice with advanced insulitis compared with 14w *Tyk2-/-* mice, but not in the PLN of normoglycemic 24w *Tyk2+/-* recipient mice (Figs. 1b and 4a). These data suggest that the defective proliferation of CD8+ T cells in the PLN of *Tyk2-/-* mice is not due to reduced IGRP epitope spreading in *Tyk2-/-* mice, but likely other factors.

NY8.3-NOD transgenic mice develop diabetes much earlier than wild type NOD mice[41]. To assess the role of TYK2 under these highly diabetogenic conditions, we analyzed the incidence of diabetes and found that it was slightly reduced in *Tyk2-/-* NY8.3 NOD mice (60%) compared with *Tyk2*-sufficient NY8.3 NOD mice (*Tyk2+/+*, 87.5%; *Tyk2+/-*, 85.7%) (Supplementary Fig. 4d). The timing of diabetes onset was comparable between the genotypes (Supplementary Fig. 4d). These results suggest that *Tyk2* deficiency is partially involved in the suppression of diabetes development even in the presence of abundant diabetogenic CD8+ T cells.

## MHC class I expression is reduced in *Tyk2*-deficient CD8⁺ resident DCs

The results of the adoptive transfer (Fig. 4a) suggest that TYK2 expression in antigen presenting cells (APCs) was associated with the efficient proliferation of islet-autoreactive CTLs in the PLN. A previous study reported the importance of TYK2 in antigen presentation by dendritic cells (DCs) for the expansion of IFN-γ⁺ CD8⁺ T cells following *Listeria monocytogenes* infection[43]. Next, we analyzed DCs, which have a central role in the adaptive immune system via antigen presentation to T cells (Supplementary Fig. 5a, b)[40]. In the PLN, the frequency of CD11c⁺ MHC II^mid resident DCs (rDCs), including CD8⁺ and CD11b⁺ subsets, and CD11c⁺ MHC II^hi migratory DCs (mDCs), including the CD103⁺ CD11b⁻, CD103⁺ CD11b⁺, and CD103⁻ CD11b⁺ subsets, were comparable irrespective of *Tyk2* genotypes (Fig. 4c and Supplementary Fig. 5c). Comparable expression levels of MHC II (I-A^g7) were observed in these DC subsets from all genotypes (Supplementary Fig. 5d). However, we found that *Tyk2* deficiency reduced the expression of MHC I (H-2K^d and H-2D^b), a crucial molecule for presenting antigens to CD8⁺ T cells[40], in CD8⁺ rDC but not other subsets of DCs (Fig. 4d and Supplementary Fig. 5e, f). This MHC I reduction in CD8⁺ rDC was not specific for the PLN as comparable results were observed in the iLN and spleen (Supplementary Fig. 5g, h). The expression levels of co-stimulatory molecules including CD40 and CD86 in CD8⁺ rDC were comparable between genotypes (Supplementary Fig. 5i). Previous studies reported that CD8⁺ rDC have a role in the expansion of CTLs by cross-priming and therefore we focused on CD8⁺ rDC[40,44]. We analyzed the gene expression profiles of CD8⁺ rDC in the PLN and found that *Interferon regulatory factor 9* (*Irf9*) and ISGs including *Ifi27l2a, Ifi44, Ifi203, and Ifih1* were highly expressed in *Tyk2⁺/⁺* CD8⁺ rDC compared with *Tyk2⁻/⁻* CD8⁺ rDC (Fig. 4e), suggesting that type I IFN signaling was upregulated in *Tyk2⁺/⁺* CD8⁺ rDC (Supplementary Fig. 5j). *Tyk2⁻/⁻* CD8⁺ rDC had higher expression levels of *Androgen receptor* (*Ar*), which might be associated with the suppression of APC functions in DCs[45], and *Il4-induced 1* (*Il4i1*), which inhibits the proliferation of T cells[46], compared with *Tyk2⁺/⁺* CD8⁺ rDC (Fig. 4e). The expression of IL-12 and IL-23, which are involved in T cell differentiation via TYK2, were comparable between genotypes.

To assess the APC function of CD8⁺ rDC, we performed an ex vivo co-culture experiment. This revealed that *Tyk2⁻/⁻* CD8⁺ rDC (pulsed with IGRP peptide or NIT-1 cells) induced lower levels of 8.3 CD8⁺ T cells expansion compared with *Tyk2⁺/⁺* CD8⁺ rDC (Fig. 4f and Supplementary Fig. 5k). These results suggest that defective TYK2-mediated type I IFN signaling in CD8⁺ rDC results in decreased MHC I expression, leading to the impaired cross-priming of islet-autoreactive CTLs in the PLN.

## Reduced T-BET⁺ CD8⁺ T cells in the PLN of *Tyk2*-deficient NOD mice

Although we found that efficient cross-priming and proliferation of CD8⁺ T cells was evident in the PLN of *Tyk2*-sufficient mice compared with *Tyk2⁻/⁻* mice (Fig. 4a), the number of CD8⁺ T cells in the PLN were comparable between the genotypes (Fig. 3a). Next, we characterized antigen-experienced CD8⁺ T cells that exhibited the CD44^hi phenotype. Because transcription factors (TFs) are key regulators of T cell effector functions, we examined TFs in CD44^hi CD8⁺ T cells in the PLN. The frequency of EOMES, BLIMP-1, and TCF1-positive CD44^hi polyclonal or IGRP CD8⁺ T cells in the PLN were comparable between genotypes (Supplementary Fig. 6a–c). We found that polyclonal and IGRP-specific CD44^hi CD8⁺ T cells positive for T-BET, an important TF for the production of IFN-γ, were decreased in the PLN of *Tyk2⁻/⁻* mice compared with *Tyk2*-sufficient mice (Fig. 5a, b), supporting the results of reduced IFN-γ-producing CD8⁺ T cells in the PLN of *Tyk2⁻/⁻* mice (Fig. 3d). The expression levels of T-BET were also reduced in CD44^hi T-BET⁺ IGRP CD8⁺ T cells in the PLN of *Tyk2⁻/⁻* mice compared with those of *Tyk2⁺/⁻* mice (Fig. 5c). Thus, these data suggest that the

development of T-BET⁺ CTLs is impaired in the PLN of *Tyk2⁻/⁻* NOD mice.

IL-12 is known to drive T-BET expression in T cells during their priming[47]. Next, to address the role of *Tyk2* in CD8⁺ T cells during priming, purified CD8⁺ T cells were stimulated ex vivo with anti-CD3/CD28 beads in the presence or absence of IL-12. We found that the loss of *Tyk2* in CD8⁺ T cells was not associated with the expansion of T-BET⁺ CD8⁺ T cells (Fig. 5d). However, T-BET expression level was impaired in *Tyk2⁻/⁻* CD8⁺ T cells compared with *Tyk2⁺/⁻* CD8⁺ T cells when the cells were stimulated with IL-12 (Fig. 5e). These data suggest that *Tyk2⁻/⁻* CD8⁺ T cells develop CTLs that are expressed lower levels of T-BET than those of *Tyk2*-sufficient CD8⁺ T cells in the PLN due to defective CD8⁺ T cell-intrinsic IL-12 signaling. In addition, the decreased number of T-BET⁺ CTLs in the PLN of *Tyk2⁻/⁻* mice might be related to defective APC functions of CD8⁺ rDC.

Chemotaxis is an important mechanism in the induction of insulitis. A previous study reported *Cxcr3*, a receptor for CXCL9 and CXCL10, is a target gene of T-BET[48]. In addition, the expression of *Cxcl10* in β-cells has an important role in the chemotaxis of islet-autoreactive T cells[49]. Next, we examined the expression of CXCR3 and found that polyclonal and IGRP-specific CD44^hi CXCR3⁺ CD8⁺ T cells were decreased in the PLN of *Tyk2⁻/⁻* mice compared with age-matched *Tyk2*-sufficient littermates (Fig. 5f, g, and Supplementary Fig. 6d, e). The decreased number of CD44^hi CXCR3⁺ CD8⁺ T cells was specific for the PLN of *Tyk2⁻/⁻* mice because there were no differences in the iLN and spleen (Supplementary Fig. 6f). In contrast, the frequency of CXCR3 expressing CD4⁺ T cells in the PLN was comparable between genotypes (Supplementary Fig. 6e). Together, these observations suggest that the comparable number of CD8⁺ T cells in the PLN between *Tyk2* genotypes (Fig. 3a) might be related to the developed CTLs expressing CXCR3, exiting the PLN, and migrating to the islets, resulting in similar numbers of CD8⁺ T cells in the PLN between the genotypes but increased numbers of CD8⁺ T cells in the pancreas of *Tyk2*-sufficient mice compared with *Tyk2⁻/⁻* mice (Fig. 3a).

## CTLs from *Tyk2*-deficient mice display impaired cytotoxicity

Our data reveal that the CD8⁺ rDC-mediated cross-priming of CD8⁺ T cells and the development of islet-autoreactive CD8⁺ T-BET⁺ CTLs are impaired but not abolished in the PLN of *Tyk2⁻/⁻* mice. It was demonstrated that stem-like CD8⁺ T cells in the PLN give rise to pathogenic CD8⁺ T cells in the pancreas with upregulating effector functions[38]. Next, to test whether the transition of gene expression profiles in islet-autoreactive CTLs depends on *Tyk2* genotypes, we performed transcriptome analysis of purified antigen-experienced CD44^hi IGRP CD8⁺ T cells in the PLN and IGRP CD8⁺ T cells in the pancreas (Fig. 5h). The expression status of CD44 and CD62L in the IGRP CD8⁺ T cells in the PLN resembled those of polyclonal CD8⁺ T cells (Supplementary Fig. 6g, h). In the pancreas, IGRP CD8⁺ T cells exhibited a CD62L⁻ phenotype (Supplementary Fig. 6h). The gene expression profiles of IGRP CD8⁺ T cells in the pancreas differed from those in the PLN[38], irrespective of the *Tyk2* genotypes (Fig. 5h). Nevertheless, we found that 744 genes were differentially expressed and that *Gzmb* and *Fasl* were upregulated in *Tyk2⁺/⁺* IGRP CD8⁺ T cells in the pancreas compared with *Tyk2⁻/⁻* cells (Fig. 5i; Group 3). In contrast, *Tyk2⁻/⁻* IGRP CD8⁺ T cells purified from the pancreas upregulated apoptosis-related genes, including *Casp3* and *Birc5*, compared with *Tyk2⁺/⁺* cells (Fig. 5i; Group 4). However, binding of annexin V, a marker of early apoptosis, in IGRP CD8⁺ T cells purified from the pancreas was comparable between genotypes (Supplementary Fig. 6i), suggesting that *Tyk2⁻/⁻* CD8⁺ T cells were not conspicuously prone to apoptosis in the pancreas. These results suggest that the loss of *Tyk2* impairs the upregulation of effector functions in islet-autoreactive CTLs in the pancreas.

A previous study reported that TYK2 had a role in CTL-mediated tumor surveillance[50]. To assess the cytotoxic activity of CTLs against

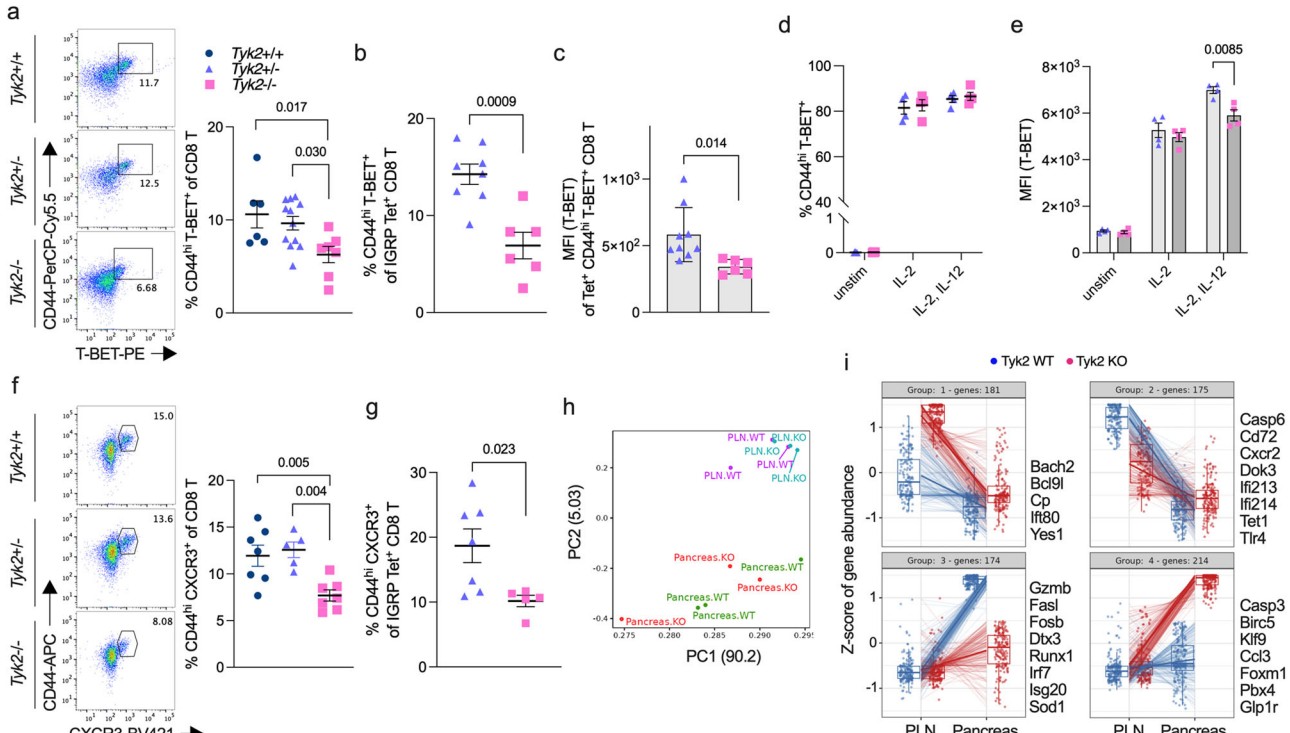

**Fig. 5 | Defective development of CD8⁺ T-BET⁺ CTLs in the PLN and effector function of CD8⁺ CTLs in the pancreas of *Tyk2*-deficient NOD mice. a** (Left) Representative flow cytometry plots and (right) graph indicate the frequency of CD44hi T-BET⁺ polyclonal cells among CD8⁺ T cells in the PLN of 6w female mice (*Tyk2⁺/⁺*, n = 6; *Tyk2⁺/⁻*, n = 12; *Tyk2⁻/⁻*, n = 7). **b** Frequency of CD44hi T-BET⁺ cells among IGRP-specific CD8⁺ T cells in the PLN of 6w female mice (*Tyk2⁺/⁻*, n = 8; *Tyk2⁻/⁻*, n = 6). **c** Protein expression levels of T-BET in CD44hi T-BET⁺ IGRP-specific CD8⁺ T cells in the PLN of 6w female mice (*Tyk2⁺/⁻*, n = 9; *Tyk2⁻/⁻*, n = 6). Frequency of CD44hi T-BET⁺ cells (**d**) and protein expression levels of T-BET (**e**) in unstimulated or ex vivo-activated *Tyk2⁺/⁻* or *Tyk2⁻/⁻* CD8⁺ T cells with anti-CD3/CD28 beads and IL-2 in the presence or absence of IL-12 for 3d (*Tyk2⁺/⁻*, n = 4; *Tyk2⁻/⁻*, n = 4). **f** (Left) Representative flow cytometry plots and (right) graph indicate the frequency of CD44hi CXCR3⁺ cells among polyclonal CD8⁺ T cells in the PLN of 6w female mice (*Tyk2⁺/⁺*, n = 7; *Tyk2⁺/⁻*, n = 5; *Tyk2⁻/⁻*, n = 7). **g** Frequency of CD44hi CXCR3⁺ cells

among IGRP-specific CD8⁺ T cells in the PLN of 6w female mice (*Tyk2⁺/⁻*, n = 7; *Tyk2⁻/⁻*, n = 5). **h** PCA of transcriptome data of purified CD44hi IGRP-specific CD8⁺ T cells in the PLN (*Tyk2⁺/⁺*, n = 3; *Tyk2⁻/⁻*, n = 3) and IGRP-specific CD8⁺ T cells in the pancreas (*Tyk2⁺/⁺*, n = 3; *Tyk2⁻/⁻*, n = 3). Gene expression profiles in the purified cells were analyzed by RNA-seq analysis. **i** Box plots show the pattern of differentially expressed genes in IGRP-specific CD8⁺ T cells in the PLN (*Tyk2⁺/⁺*, n = 3; *Tyk2⁻/⁻*, n = 3) and pancreas (*Tyk2⁺/⁺*, n = 3; *Tyk2⁻/⁻*, n = 3). Likelihood ratio tests were used to determine differentially expressed genes between the genotypes (FDR < 0.1, fold change >1.5). Selected genes are highlighted on the right side of each box plots. Boxes and whiskers represent the median and the lower and upper hinges correspond to the 25th and 75th percentiles. The whiskers extend from the hinges to the values no further than 1.5 times the inter-quartile ranges from each box hinge. Data represent the mean ± SEM. *P*-values were calculated using one-way ANOVA with Dunnett's posttest (**a**, **f**) and two-tailed Student's *t* test (**b**–**e**, and **g**).

β-cells, ex vivo-activated 8.3 CD8⁺ T cells in the presence of IL-2 and IL-12 (8.3 CTLs) were cultured with NIT-1 cells, a pancreatic β-cell line derived from female NOD mice[51], and the expression levels of effector molecules in the CTLs and the survival of NIT-1 cells were analyzed[52]. After priming of the 8.3 CTLs, the expression levels of GZMB in the CTLs were comparable between genotypes (Fig. 6a). However, the 8.3 CTLs cultured with NIT-1 cells had upregulated expression levels of GZMB, which were attenuated by *Tyk2* deficiency, blockade of interferon alpha and beta receptor subunit 1 (IFNAR1), and neutralization of IFN-β, but not IFN-α (Fig. 6a, b). The expression levels of *Ifnb1* in NIT-1 cells were upregulated after coculturing with 8.3 CTLs (Fig. 6c). These results suggest that TYK2-mediated IFN-β signaling in CTLs is involved in the enhancement of the expression levels of GZMB in the CTLs after target cell encounter. Indeed, ISGs, including *Irf7* and *Isg20*, were upregulated in *Tyk2⁺/⁺* IGRP CD8⁺ T cells in the pancreas compared with *Tyk2⁻/⁻* IGRP CD8⁺ T cells (Fig. 5i; Group 3). The expression levels of FASL and CASP3/7 in the 8.3 CTLs after NIT-1 cell encounter were comparable between genotypes (Supplementary Fig. 6j). The cytotoxic function of *Tyk2⁻/⁻* 8.3 CTLs primed with IL-12 had a reduced killing capacity compared with those of *Tyk2⁺/⁻* 8.3 CTLs (Fig. 6d). Together, these results suggest that defective CTL-intrinsic IFN-β signaling related to *Tyk2* deficiency reduces the cytotoxic activity of CTLs after encountering β-cells.

## TYK2 inhibitor has the potential to prevent the development of autoimmune T1D

Recently, BMS-986165, a selective TYK2 inhibitor[53], was reported to inhibit autoimmune diseases including psoriasis in mice and humans[21,54], although its effects on autoimmune T1D are unknown. Our results suggest that TYK2 is a potential target for preventing autoimmune T1D. To test whether BMS-986165 inhibited the development of CTLs, purified *Tyk2⁺/⁺* CD8⁺ T cells were stimulated ex vivo with anti-CD3/CD28 beads, IL-2, and IL-12, in the presence or absence of BMS-986165. The expression levels of T-BET in stimulated CD8⁺ T cells were reduced BMS-986165 dose-dependently (Fig. 7a, and Supplementary Fig. 7a). Consistent with the T-BET expression levels, the β-cell killing capacity of stimulated 8.3 CD8⁺ T cells was impaired BMS-986165 dose-dependently (Fig. 7b), supporting the results of reduced killing capacity in *Tyk2⁻/⁻* 8.3 CD8⁺ T cells (Fig. 6d). Next, we assessed CXCR3 expression levels in the stimulated CD8⁺ T cells. Re-culture of stimulated CD8⁺ T cells for 2d induced CXCR3 expression in the cells (Fig. 7c)[55]. However, re-cultured CD8⁺ T cells primed with BMS-986165 had reduced CXCR3 expression levels dose-dependently (Fig. 7c), suggesting that the reduction of CXCR3 in CTLs was due to a CTL-intrinsic mechanism. Next, to test the effect of BMS-986165 on developed CTLs, we treated ex vivo stimulated CD8⁺ T cells with BMS-986165 for 2d and

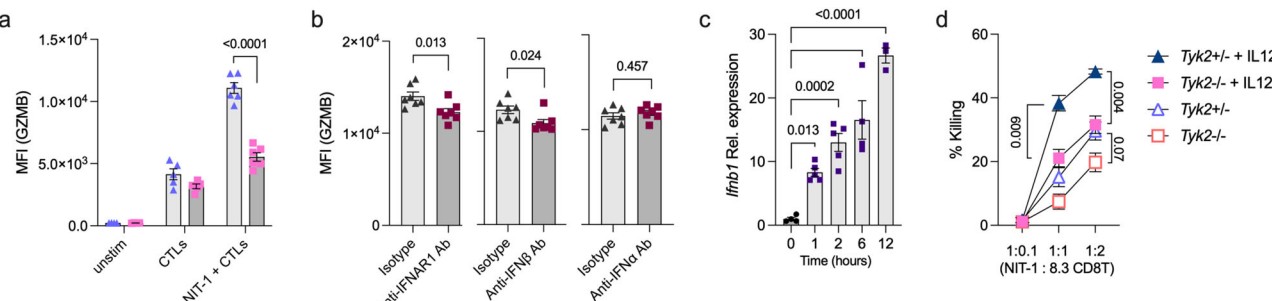

**Fig. 6 | Loss of *Tyk2* impairs the cytotoxic activity of CD8⁺ CTLs. a** Protein expression levels of GZMB in the unstimulated 8.3 CD8⁺ T cells, ex vivo-activated 8.3 CD8⁺ T cells, or ex vivo-activated 8.3 CD8⁺ T cells cocultured with NIT-1 cells (*Tyk2*⁺/⁻, n = 6, *Tyk2*⁻/⁻, n = 6). Purified 8.3 CD8⁺ T cells were activated with IL-2, IL-12, and anti-CD3/28 beads for 3d. For coculture, ex vivo-activated 8.3 CD8⁺ T cells were cultured with NIT-1 cells for 24 h (CTLs:NIT-1 = 1:1). **b** Protein expression levels of GZMB in *Tyk2*⁺/⁻ 8.3 CTLs after 24 h of coculture with NIT-1 cells in the presence of IFNAR1 blocking antibody (n = 7), IFN-β neutralizing antibody (n = 7), or IFN-α neutralizing antibody (n = 7). Purified *Tyk2*⁺/⁻ 8.3 CD8⁺ T cells were activated with IL-

2, IL-12, and anti-CD3/28 beads for 3d, and cultured with NIT-1 cells. **c** Gene expression levels of *Ifnb1* in NIT-1 cells after coculture with ex vivo-activated 8.3 CD8⁺ T cells (0 h, n = 4; 1 h, n = 5; 2 h, n = 5; 6 h, n = 4; 12 h, n = 3). **d** The killing capacity of ex vivo-activated 8.3 CTLs against NIT-1 cells at the indicated ratios. Purified 8.3 CD8⁺ T cells were activated with IL-2 and CD3/28 beads in the presence or absence of IL-12 for 3d. NIT-1 cells were cultured with activated 8.3 CD8⁺ T cells for 24 h, and then the survival of NIT-1 cells was analyzed (1:0.1, n = 4; 1:1, n = 3; 1:2, n = 3). Data represent the mean ± SEM. *P*-values were calculated using one-way ANOVA with Dunnett's posttest (**c**) and two-tailed Student's *t* test (**a**, **b**, and **d**).

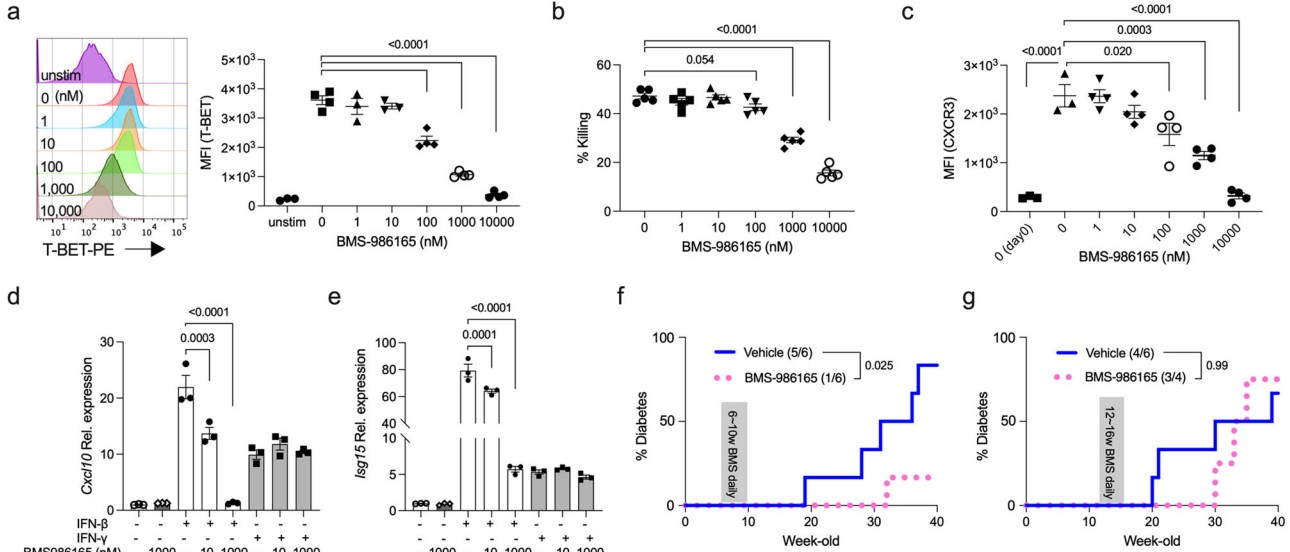

**Fig. 7 | BMS-986165 prevents the development of autoimmune T1D. a** (Left) Representative histogram and (right) protein expression levels of T-BET in stimulated *Tyk2*⁺/⁺ CD8⁺ T cells. Purified *Tyk2*⁺/⁺ CD8⁺ T cells were activated with IL-2, IL-12, and anti-CD3/28 beads for 3d in the presence or absence of BMS-986165 (unstimulated control, n = 3; stimulated cells, n = 4). **b** The β-cell killing capacity of ex vivo-activated *Tyk2*⁺/⁺ 8.3 CD8⁺ T cells. Purified 8.3 CD8⁺ T cells were primed with IL-2, IL-12, and anti-CD3/28 beads for 3d in the presence or absence of BMS-986165. NIT-1 cells were cultured with activated 8.3 CD8⁺ T cells for 24 h, and then the survival of NIT-1 cells was analyzed (n = 5). **c** Protein expression levels of CXCR3 in stimulated *Tyk2*⁺/⁺ CD8⁺ T cells. Ex vivo activated CD8⁺ T cells with IL-2, IL-12, and anti-CD3/28 beads for 3d in the presence or absence of BMS-986165 were re-cultured without BMS-986165 for 48 h and the induction of CXCR3 was analyzed

(unstimulated control and stimulated cells without BMS-986165, n = 3; stimulated cells with BMS-986165, n = 4). Day 0 data indicate the CXCR3 expression levels just after ex vivo stimulation. Gene expression levels of *Cxcl10* (**d**) and *Isg15* (**e**) in NIT-1 cells stimulated with 100 U/mL of IFN-β or 0.1 ng/mL of IFN-γ in the presence or absence of BMS-986165 for 3 h (n = 3). **f** Incidence of diabetes in vehicle (n = 6) or BMS-986165-treated (n = 6) wild type female NOD mice for 4w from 6w of age. Mice were dosed by oral gavage once daily with BMS-986165 30 mg/kg in vehicle (EtOH:TPGS:PEG300, 5:5:90) or vehicle alone. Diabetes was defined by a non-fasting blood glucose level exceeding 250 mg/dL. **g** Incidence of diabetes in vehicle (n = 6) or BMS-986165-treated (n = 4) wild type female NOD mice for 4w from 12w of age. Data represent the mean ± SEM. *P*-values were calculated using one-way ANOVA with Dunnett's posttest (**a**–**e**) and log-rank test (**f**, **g**).

found that the expression levels of T-BET and CXCR3 were unchanged (Supplementary Fig. 7b). However, BMS-986165 inhibited IL-12 and IL-18-induced bystander IFN-γ production in CD44ʰⁱ CD8⁺ T cells dose-dependently (Supplementary Fig. 7c). These results suggest that TYK2 has a role in the development of CD8⁺ T-BET⁺ CXCR3⁺ CTLs and bystander activation of CD8⁺ CTLs upon cytokines stimulation, and is dispensable for the maintenance of T-BET and CXCR3 expression in CD8⁺ CTLs.

Next, to test whether BMS-986165 inhibited inflammatory responses in β-cells, NIT-1 cells with or without BMS-986165 were stimulated with type I or type II IFNs. BMS-986165 inhibited IFN-β-induced *Cxcl10*, *Isg15*, and *Oas2* expressions dose-dependently (Fig. 7d, e, and Supplementary Fig. 7e). IFN-β-induced *Cxcl9*, *Cxcl13*, and *Fas* expression levels were reduced but not significantly suppressed by BMS-986165 treatment (Supplementary Fig. 7e). BMS-986165 had no effect on IFN-γ-induced these genes expression levels in

NIT-1 cells, because TYK2 is not associated with type II IFN signaling, confirming the high selectively of the TYK2 inhibitor. Together, these results demonstrate that BMS-986165 inhibits the development and bystander activation of CD8$^+$ CTLs, and type I IFN-induced gene expressions in β-cells.

To demonstrate the effects of BMS-986165 on the development of autoimmune T1D in vivo, we administrated the inhibitor to wild type NOD mice by oral gavage once-daily at 30 mg/kg for 4 weeks from 6w of age. NOD mice treated with BMS-986165 had reduced diabetes incidence (1/6) compared with vehicle control mice (5/6) ($p = 0.025$, Fig. 7f). The timing of diabetes onset was delayed in the inhibitor treated mice (32w) compared with vehicle treated mice (19w) (Fig. 7f). In contrast, treatment with BMS-986165 at the same dose from 12w of age resulted in comparable diabetes incidence (3/4) with vehicle control mice (4/6) ($p = 0.99$, Fig. 7g). However, the timing of diabetes onset was delayed in the inhibitor treated mice (30w) compared with vehicle treated mice (20w) (Fig. 7g). These results suggest that treatment with a TYK2 inhibitor started at an early stage of T1D may be a potential strategy to prevent the development of autoimmune T1D.

## Discussion

In the context of autoimmune disease, the role of TYK2 in CD4$^+$ T cells, but not CD8$^+$ T cells, has been well described[36,56]. Our study reveals that IL-12 signaling via TYK2 upregulated T-BET expression in CD8$^+$ CTLs during their priming, leading to the upregulation of CXCR3. In addition, the efficient cross-priming of CTLs by CD8$^+$ rDC may be associated with MHC I expression levels in CD8$^+$ rDC, probably induced by type I IFN signaling via TYK2. These data provide evidence that TYK2 has a critical role in the generation of autoreactive CD8$^+$ CTLs. However, increased naïve CD8$^+$ T cells and decreased IFN-γ-producing CD8$^+$ T cells in the PLN of $Tyk2^{-/-}$ mice were limited in the initiation phase of T1D (Fig. 3b, d) when the PLN is dispensable for T1D development[39]. This is likely because loss or inhibition of TYK2 did not completely inhibit the development of T-BET$^+$ CTLs (Figs. 5a and 7a), the number of CTLs in the PLN of $Tyk2^{-/-}$ mice became comparable to that in the PLN of $Tyk2$-sufficient mice in accordance with aging. In addition to the impaired development of CD8$^+$ T-BET$^+$ CTLs in the PLN of $Tyk2^{-/-}$ mice, the reduced killing capacity of $Tyk2^{-/-}$ CD8$^+$ CTLs against β-cells probably contributes to the suppression of the development of autoimmune T1D. Importantly, TYK2-mediated IFN-β signaling was associated with the upregulation of effector molecule GZMB in CD8$^+$ CTLs after β-cell encounter (Fig. 6a, b). Thus, TYK2 is involved in the development of CD8$^+$ T-BET$^+$ CTLs via IL-12 signaling and the induction of effector functions via type I IFN signaling after encountering target cells.

Consistent with an earlier study[57], a similar frequency of Treg was observed regardless of the $Tyk2$ genotypes (Supplementary Fig. 3h). The non-involvement of TYK2 in Treg differentiation and maintenance may have led to the efficient suppression of autoreactive CTLs in $Tyk2^{-/-}$ mice[57]. Although the differentiation of Th1 and Th17 cells in the PLN was comparable between $Tyk2$ genotypes, bystander IFN-γ production by CD44$^{hi}$ CD4$^+$ T cells was impaired by TYK2 inhibition (Supplementary Fig. 7d), suggesting the pathogenic role of autoreactive CD4$^+$ T cells may be also defective in $Tyk2^{-/-}$ mice.

Our data suggest that TYK2 signaling in β-cells becomes active with age. Because the expression levels of chemokines, ISGs, and MHC I molecules were lower in the β-cells from 6w mice compared with those from 11w mice (Fig. 2d, e), the increased inflammatory signature of β-cells with age may be caused by infiltrating immune cells or the local islet environment induced by immune cells. We found that $Cxcl9$ and $Cxcl10$ expressions in β-cells were upregulated with age, which were slightly attenuated by $Tyk2$ deficiency, at least in part, due to impaired type I IFN signaling in β-cells (Fig. 7d, e, and Supplementary Fig. 7e). In addition to the low $Cxcl9$ and $Cxcl10$ expressions, reduced MHC I and $Fas$ expressions in the β-cells of $Tyk2^{-/-}$ mice may contribute

to the preservation of β-cells from CTLs attack. In the early phase of T1D development, TYK2 inhibition might reduce the expression levels of islet-associated autoantigens in β-cells (Supplementary Fig. 1g). In contrast to these preservation effects of TYK2 inhibition in β-cells, a recent study suggested that TYK2 inhibition lead to reduced β-cell mass related to the defective formation of pancreatic endocrine precursors[35]. Consequently, TYK2 may has a different role in β-cells depending on the stage of T1D and/or developmental stage of β-cells.

Together, these findings suggested that the mechanisms described above act in concert to inhibit the development of autoimmune T1D in $Tyk2^{-/-}$ NOD mice. In support of this, a recent study reported that the loss-of-function variant rs34536443 in $TYK2$ was associated with a lower risk of autoimmune disease including T1D[23]. Thus, TYK2 is involved in the pathogenesis of autoimmune T1D in mice and humans.

Treatment with the TYK2 inhibitor BMS-986165 impaired the development of CD8$^+$ T-BET$^+$ CXCR3$^+$ CTLs, bystander activation of CD8$^+$ CTLs, and chemokine and ISG expressions in β-cells dose-dependently. Of note, a reduced T1D progression rate was evident when the treatment was started at 6w but not at 12w of age, suggesting that the initiation or early phase of T1D is the therapeutic window for the TYK2 inhibitor. The abundant presence of IFN-γ-producing CD8$^+$ T cells in the PLN during the initiation phase of T1D (Fig. 3d) may explain why intervention in the initiation phase of T1D is optimal for the efficient suppression of T1D development. Even though the overall incidence of diabetes was not reduced, mice treated with the TYK2 inhibitor from 12w had a delayed onset of T1D compared with vehicle treated mice (Fig. 7g). These observations suggest that early intervention with a TYK2 inhibitor may be a potent immunotherapeutic strategy for autoimmune T1D, and treatment with the inhibitor in the late stage of T1D may also be effective at suppressing the autoimmune process. The effect of the TYK2 inhibitor on autoimmune T1D will be better defined by studies using large group sizes. In addition, because our conclusions were based on experiments using female mice, further studies with male mice will be needed.

TYK2 is involved in anti-microorganism responses via cytokine signaling[11,12,43]. Clinical studies reported potential adverse effects of BMS-986165, including nasopharyngitis, diarrhea, nausea, and upper respiratory tract infection[23,26]. In the clinical trial, BMS-986165 was considered safe on the basis of observations of a slight increase but low rate of serious viral infection[26]. However, it was reported that some patients with T1D, especially with acute onset and fulminant T1D, exhibited severe or mild flu-like syndromes at disease onset[17,58–62]. In contrast to the role of TYK2 in autoimmune T1D described in this paper, we reported that reduced TYK2 expression was a risk factor for β-cell tropic virus-induced diabetes and impaired insulin secretion[11,17,63]. Of note, the importance of anti-viral defense in β-cells to prevent β-cell destruction caused by viruses was reported based on in vivo findings[11,32,64,65]. Our studies suggest that the role of TYK2 in T1D depends on causal factors. In contrast, TYK2 inhibition reduces inflammation in β-cells in response to cytokines and a mimic of viral RNA, leading to the preservation of β-cells[19,35]. Although the role of viruses in T1D is controversial, the advantages and risks of immunomodulation should be considered when developing treatments for T1D. An anti-diabetogenic virus vaccine, which is currently under development[66], might be a potent option to eliminate the potential viral contribution to β-cell destruction in patients taking immunosuppressant drugs such as TYK2 inhibitors and anti-CD3 antibodies, regardless of the mechanisms of T1D development induced by viruses. In addition to the increased infection risk, there is a potential risk of lung cancer, non-Hodgkin lymphoma, and reduced β-cell mass with TYK2 inhibition[35,67], indicating that further safety assessments will be needed for long-term treatment with TYK2 inhibitors.

In conclusion, our study demonstrated that the blockade of TYK2 signaling in autoimmune T1D model NOD mice impaired the development and effector functions of CD8$^+$ CTLs, the efficient

cross-priming of CD8[+] CTLs by CD8[+] rDC, and the expression of inflammatory molecules in β-cells, leading to the inhibition of the onset of overt autoimmune T1D. The evaluation of candidate genes by multifaceted approaches will aid our understanding of the complex role of genes involved in multifactorial diseases and help the development of the safe and effective preventive/therapeutic strategies.

## Methods

### Animals and cells

Wild type NOD/shiJcl mice were obtained from Clea Japan, Inc. (Tokyo, Japan) and housed in our facility. Female mice were used for all experiments. *Tyk2* KO mice on a C57BL/6J genetic background were generated by backcrossing 129/Sv-derived Tyk2 KO mice to C57BL/6J mice for 12 times[11,30]. *Tyk2*KO mice on a NOD/shiJcl genetic background (*Tyk2*KO.NOD) were generated by backcrossing with NOD/shiJcl mice more than ten times. The STR loci were analyzed by ICLAS Monitoring Center (Kanagawa, Japan). The microsatellite repeat polymorphisms at the Idd loci were tested by PCR[27]. *Tyk2* heterozygous and homozygote mice were healthy and fertile. NOD.SCID mice were obtained from the Jackson Laboratory Japan, Inc. (RRID:IMSR_JAX:001303) (Yokohama, Japan) and bred in our facility. *Tyk2*KO mice on a NOD.SCID genetic background (*Tyk2*KO.NOD.SCID) were generated by crossing with *Tyk2*KO.NOD mice. The *Prkdcscid* gene mutation was determined by PCR[68]. NY8.3 NOD mice were provided by S. Akazawa and N. Abiru (Nagasaki University, Nagasaki, Japan) (RRID:IMSR_JAX:005868)[41]. *Tyk2*KO mice on a NY8.3 NOD genetic background (*Tyk2*KO 8.3 NOD) were generated by crossing with *Tyk2*KO.NOD mice. Blood glucose levels were measured in tail vein blood using Glutest Ai (Sanwa, Nagoya, Japan). Diabetes was defined by a non-fasting blood glucose level exceeding 250 mg/dL. All mice were maintained on a 12-hour light/dark cycle in a temperature-controlled facility under specific pathogen-free conditions at $23 \pm 2\,°C$ with $50 \pm 10\%$ humidity, and provided with sterile food and water *ad libitum*. SCID mice were fed with sterile water. Experimental and control animals were housed in co-housed condition. All animals were euthanized by cervical dislocation under isoflurane anesthesia. This study was approved by the Saga University Animal Care and Use Committee and conducted in accordance with the regulations on animal experimentation at Saga University.

NIT-1 cells, a pancreatic β-cell line derived from female NOD mice[51], were obtained from ATCC (Manassas, VA, USA). The cells were cultured and maintained in F-12K medium (Gibco) containing 10% fetal bovine serum and 1% penicillin and streptomycin (Wako) in 5% $CO_2$ at 37 °C.

### Histology

Pancreas specimens were immersed in 10% (w/v) formaldehyde overnight at 4 °C and then embedded in paraffin. Cut sections (3 μm) were subjected to hematoxylin and eosin (H&E) staining. Insulitis scores were defined as follows: 0, no insulitis; 1, peri-insulitis; 2, infiltrative insulitis less than 50% of the islet area; and 3, infiltrative insulitis more than 50% of the islet area (Supplementary Fig. 1d).

### ELISA

IAA levels in serum were measured using an ELISA kit (Cusabio, Huston, TX, USA) according to the manufacturer's instructions. The optical density was determined at 450 nm.

### Adoptive transfer experiments

Donor splenocytes were purified from the spleens of non-diabetic 14w or 24w female mice. Splenocytes ($1\times10^7$) were intravenously transferred into 6–8w wild type or *Tyk2*KO.NOD.SCID female recipient mice. Blood glucose levels were measured in tail vein blood once weekly using Glutest Ai (Sanwa). To analyze the proliferation of CD8[+] T cells in LNs, naïve 8.3 CD8[+] T cells from the LNs of non-diabetic female NY8.3 NOD mice were purified using two rounds of the naïve CD8[+] T cell isolation kit (Miltenyi Biotec, Bergisch Gladbach, Germany) with LS columns (Miltenyi Biotec). The average purity of the sorted cells was 96%. Purified naïve 8.3 CD8[+] T cells were labeled using a CellTrace violet cell proliferation kit (Invitrogen, Waltham, MA, USA) according to the manufacturer's protocol. Then, $4 \times 10^6$ labeled naïve 8.3 CD8[+] T cells were transferred into 6–9w or 24w recipient mice. Next, 5 or 10d after the transfer, the PLN and iLN were harvested and the proliferation of CTV labeled cells were analyzed using a FACSVerse flow cytometer (BD Biosciences).

### Islets isolation and β-cell sorting

Islets were isolated using 1 mg/ml collagenase-P (Roche) and Histopaque-1077 (Sigma-Aldrich, Tokyo, Japan) as previously described with minor modifications[69]. Purification of β-cells was performed as previously described[31,32]. Briefly, isolated islets were dissociated using TrypLE express (Gibco BRL, Grand Island, NY, USA) for 9 min at 37 °C with vortexing every 3 min. Dissociated islet cells were washed with RPMI-1640 containing 10% FBS and 1% PcSM, and resuspended in the medium. Dissociated islet cells were stained with FluoZin-3-AM (Invitrogen), TMRE (Life Technologies, Carlsbad, CA, USA), anti-CD45 antibody (BioLegend, San Diego, CA, USA), and anti- MHC class I antibody (H-2K$^d$ or H-2D$^b$) (BioLegend) for 30 min at 37 °C. Then, 1 mg/mL propidium iodide (Sigma-Aldrich) was added to the cell suspension just before sorting to exclude dead cells. FluoZin-3-AM-positive, TMRE-positive, and CD45-negative cells were sorted using MA900 (Sony, Tokyo, Japan), and data were analyzed using FlowJo software (Tree Star, Ashland, OR, USA).

### Microarray

We used an Affymetrix Clariom D Assay Mouse Array for the β-cell transcriptome analysis. To identify differentially-expressed genes, we calculated the Z scores and ratios (non-log-scale fold-change) from the normalized signal intensities for each probe to compare control and experimental samples. Criteria used for identifying differentially expressed genes were as follows: upregulated genes, Z score >2.0 and ratio <1.5; downregulated genes, Z score <−2.0 and ratio >0.66. The accession number for the data is GEO: GSE235670. Microarray analysis was supported by Cell Innovator (Fukuoka, Japan).

### Flow cytometry analysis

To isolate immune cells from the LNs, thymus, or spleen, tissues were minced and enzymatically dissociated with 1 mg/mL collagenase D (Roche Applied Science) and 25 U/mL DNase I (Fujifilm Wako, Osaka, Japan) in DMEM (Fujifilm Wako) for 30 min at 37 °C. The digested tissues were filtered through a 70-μm cell strainer with a syringe plunger. Red blood cell lysis was performed by incubation with ammonium-chloride-potassium lysis buffer for 4 min on ice. Isolated cells were preincubated with CD16/32 blocker (clone 93; BioLegend) for 5 minutes at 4 °C to prevent nonspecific staining, and stained for 20 min at 4 °C with antibodies. We added 1 mg/mL propidium iodide to the cell suspension to exclude dead cells.

To isolate immune cells from the pancreas, pancreas was perfused with 1 mg/mL collagenase D, 0.5 mg/mL trypsin inhibitor (Fujifilm Wako), and 25 U/mL DNase I (Fujifilm Wako) through the bile duct. The perfused pancreas was removed carefully from the intestine, stomach, spleen, and LNs, and dissociated for 30 min at 37 °C. Then, 5 mL of DMEM containing 10% FBS and 1% PcSM was added to the dissociated pancreas and shaken for 10 s to completely dissociate the tissue. The digested pancreas was filtered through a 70-μm cell strainer with a syringe plunger. Red blood cell lysis was performed by incubation with ammonium-chloride-potassium lysis buffer for 4 min on ice. The digested pancreas was centrifuged and resuspended in 40% Percoll (GE Healthcare Life Science, Buckinghamshire, UK) layered on 70% Percoll, and centrifuged at 70 g for 20 min. Cells were collected from

the Percoll interface. Isolated cells were preincubated with CD16/32 blocker (BioLegend) for 5 min at 4 °C, and stained for 20 min at 4 °C with antibodies. We added 1 mg/mL propidium iodide to the cell suspension to exclude dead cells (Supplementary Fig. 2c).

To stain IFN-γ and IL-17A, cells were stimulated with 25 ng/mL PMA (Fujifilm Wako) and 1 mg/mL ionomycin (Fujifilm Wako) for 6 h at 37 °C. Then, 10 mg/mL Brefeldin A (Fujifilm Wako) was added for the last 5 h of incubation. Intracellular staining was performed using a Cytofix/Cytoperm Fixation/Permeabilization Solution kit (BD Biosciences, San Jose, CA, USA) according to the manufacturer's instructions. The intracellular staining of FOXP3, Ki67, T-BET, EOMES, TCF-1, GZMB, and BLIMP-1, was performed using the Foxp3/Transcription factor staining buffer set (Invitrogen) according to the manufacturer's instructions. Dead cells were excluded using a Zombie Aqua fixable viability kit (BioLegend). AnnexinV staining was performed with Annexin V binding buffer (BioLegend) according to the manufacturer's instructions. Stained cells were analyzed by FACSVerse flow cytometer (BD Biosciences) and data were analyzed using FlowJo software (Tree Star).

The following antibodies were used in this study. Details of the antibody information are shown in Supplementary Table 1. FITC anti-CD8a (53-6.7), FITC anti-CD62L (MEL-14), FITC anti-CD4 (RM4-5), FITC anti- RT1B (I-A g7) (OX-6), PE anti-CD4 (GK1.5), PE anti-CD3 (17A2), PE anti-FOXP3 (MF-14), PE anti-IFN-γ (XMG1.2), PE anti-KI67 (16A8), PE anti-T-bet (4B10), PE anti-BLIMP-1 (5E7), PE anti-CD40 (3/23), PE Annexin V, PerCP Cy5.5 anti-CD3 (17A2), PerCP Cy5.5 anti-GZMB (QA16A02), PerCP Cy5.5 anti-CD44 (1M7), APC anti-CD19 (1D3), APC anti-CD44 (1M7), APC anti-CD25 (PC61), APC anti-Cxcr3 (CXCR3-173), APC anti-CD86 (GL-1), APC anti-CD103 (2E7), APC anti-H-2D$^b$ (KH95), APC-Cy7 anti-CD3 (17A2), APC-Cy7 anti-TCR γ/δ (GL3), APC-Cy7 anti-CD45 (30-F11), APC-Cy7 anti-CD8a (53-6.7), PE-Cy7 anti-TCR γ/δ (GL3), PE-Cy7 anti-CD8a (53-6.7), PE-Cy7 anti-CD4 (RM4-5), PE-Cy7 anti-PD1 (RMP1-30), BV421 anti-H2-Kd (SF-1.1), BV421 anti-Cxcr3 (CXCR3-173), BV421 anti-CD103 (2E7), and BV510 anti-CD45 (30-F11) were purchased from BioLegend. PE anti-CD3 (500A2), PE anti-CD11b (M1/70), PerCP Cy5.5 anti-CD19 (1D3), PE anti-CD44 (1M7), PE-Cy7 anti-EOMES (Dan11mag), and PE-Cy7 anti-CD11c (N418) were purchased from eBioscience. BV450 anti-CD4 (RM4-5), APC anti-IL-17A (TC11-18H10), and BV450 anti-CD11b (M1/70) were purchased from BD Biosciences. PE anti-TCF1 (C63D9) was purchased from Cell Signaling Technology.

## Quantitative PCR

mRNA was extracted using Isogen II (Nippon gene, Tokyo, Japan) and cDNA was synthesized using a High-capacity cDNA reverse transcription kit (Applied Biosystems, Foster City, CA, USA) according to the manufacturer's instructions. PCR was performed on a PCR thermal cycler (Takara, Tokyo, Japan) and real-time PCR was performed using QuantStudio 3 (Thermo Fisher Scientific, Waltham, MA, USA). The relative quantification value is expressed as $2^{-\Delta\Delta Ct}$, where $\Delta\Delta Ct$ is the difference between the mean $Ct$ value of duplicate measurements of the sample and the endogenous *Actb* control. *Actb*: primer set ID, MA050368 (Takara Bio); *Cxcl10*: primer set ID, MA118556 (Takara Bio); *Cxcl9*: forward, 5′-CCTAGTGATAAGGAATGCACGATG-3′, reverse, 5′-CTAGGCAGGTTTGATCTCCGTTC-3′; *Cxcl13*: forward, 5′-CATAGATCGGATTCAAGTTACGCC-3′, reverse, 5′-GTAACCATTTGGCACGAGGATTC-3′; *Fas*: forward, 5′-CTGCGATTCTCCTGGCTGTGAA-3′, reverse, 5′-CAACAACCATAGGCGATTTCTGG-3′; *Isg15*: forward, 5′-CATCCTGGT-GAGGAACGAAAGG-3′, reverse, 5′-CTCAGCCAGAACTGGTCTTCGT-3′; *Oas2*: forward, 5′-CCTTGGAAAGTGCCAGTACCTA-3′, reverse, 5′-CCTTGGTCCTGCCCACAAGAT-3′, and *Ifnb1*: forward, 5′-GCCTTTGCC ATCCAAGAGATGC-3′, reverse, 5′-ACACTGTCTGCTGGTGGAGTTC-3′.

## RNA-sequencing analysis

To analyze CD44$^{hi}$ IGRP CD8$^+$ T cells and CD8$^+$ rDC in the PLN, NOD mice aged 6w were used. Approximately 300–600 cells were sorted

and used for the analysis. To analyze IGRP CD8$^+$ T cells in the pancreas, NOD mice aged 10–12w were used. Approximately 600 cells were sorted and used for the analysis. Complementary DNA was prepared using a SMART-Seq v4 Ultra Low Input RNA Kit for Sequencing (Clontech Laboratories, CA, USA). Libraries were prepared using a Nextera XT DNA Library Prep Kit (Illumine, San Diego, CA, USA). Samples were run on a NovaSeq 6000 (Illumina). An average of 64 million paired reads were generated per sample. The accession number for the data is SRA: PRJNA992677. RNA-seq analysis was supported by Takara Bio Inc. (Japan).

## Ex vivo co-culture assay

To obtain FACS-sorted CD8$^+$ rDCs, spleens, iLN, and PLN from non-diabetic 6w–8w female mice were minced and enzymatically dissociated with 1 mg/mL collagenase D (Roche Applied Science) and 25 U/mL DNase I in DMEM for 30 min at 37 °C, as described above. The digested tissues were filtered through a 70-μm cell strainer with a syringe plunger. Red blood cell lysis was performed by incubation with ammonium-chloride-potassium lysis buffer for 5 min on ice. The digested tissues were centrifuged and resuspended in 40% Percoll (GE Healthcare Life Science) layered on 70% Percoll, and centrifuged at 70 g for 20 min. Cells were collected from the Percoll interface. Then, isolated cells were preincubated with CD16/32 blocker (BioLegend) for 5 minutes at 4 °C, and stained for 5 min at 4 °C with biotinylated anti-CD3 antibody (17A2; BioLegend) to exclude CD3-positive T cells in the cell suspension using anti-biotin microbeads (Miltenyi Biotec) and LS columns (Miltenyi Biotec). Next, the cells were stained for 5 min at 4 °C with biotinylated anti-CD19 antibody (1D3; BioLegend) to exclude CD19-positive B cells in the cell suspension using anti-biotin microbeads and LS columns. The cells were stained with antibodies (FITC anti-RT1B (OX-6), PE anti-CD11b (M1/70), APC anti-CD103 (2E7), APC-Cy7 anti-CD8α (53-6.7), PE-Cy7 anti-CD11c (N418), BV510 anti-CD45 (30-F11)) and sorted using MA900 (Sony). To obtain naïve 8.3 CD8$^+$ T cells, naïve CD8$^+$ T cells were isolated from the LNs of NY 8.3 NOD mice using two rounds of the naïve 8.3 CD8$^+$ T cell isolation kit (Miltenyi Biotec) with LS columns (Miltenyi Biotec), following the manufacturer's instructions. Purified naïve 8.3 CD8$^+$ T cells were labeled using a CellTrace violet cell proliferation kit (Invitrogen) according to the manufacturer's protocol.

For the co-culture assay (naïve CD8$^+$ T, CD8$^+$ rDC, and IGRP peptide), $2 \times 10^4$ sorted CD8$^+$ rDC and $1 \times 10^5$ CTV labeled naïve 8.3 CD8$^+$ T cells (DC:CD8$^+$ T ratio = 1:5) were mixed and cultured in U-bottom 96-well plates in RPMI-1640 with 10% FBS, 1% PcSM, 50 μM 2-mercaptoethanol, and 1 ng/mL IGRP$_{206-214}$ (VYLKTNVFL). Cultures of CTV-labeled naïve 8.3 CD8$^+$ T cells and 1 ng/mL IGRP$_{206-214}$ peptide without CD8$^+$ rDC for 3d were used as a control.

For co-culture assays with NIT-1 cells (naïve CD8$^+$ T, CD8$^+$ rDC, and NIT-1 cells without IGRP peptide), $2 \times 10^4$ sorted CD8$^+$ rDC and $1 \times 10^5$ CTV labeled naïve 8.3 CD8$^+$ T cells (DC:CD8$^+$ T ratio = 1:5) were mixed and cultured in U-bottom 96-well plates in RPMI-1640 with 10% FBS, 1% PcSM, 50 μM 2-mercaptoethanol, and $1 \times 10^4$ NIT-1 cells. Cultures of CTV-labeled naïve 8.3 CD8$^+$ T cells and NIT-1 cells without CD8$^+$ rDC for 3d were used as controls. The cells were cultured at 37 °C and 5% CO$_2$ for 72 h and then the dye dilution was analyzed using a FACSVerse flow cytometer (BD Biosciences), and data were analyzed using FlowJo software (Tree Star).

## Ex vivo cytotoxicity assay

To assess the effector functions of CD8$^+$ T cells against β-cells, ex vivo activated 8.3 CD8$^+$ T cells and NIT-1 cells were co-cultured as previously described[52]. Briefly, 8.3 CD8$^+$ T cells were isolated from the LNs of non-diabetic NY 8.3 NOD female mice using two rounds of the CD8$^+$ T cell isolation kit (Miltenyi Biotec) with LS columns (Miltenyi Biotec), following the manufacturer's instructions. The isolated cells were cultured in U-bottom 96-well plates in RPMI-1640 with 10% FBS, 1% PcSM,

50 µM 2-mercaptoethanol, 30 U/mL IL-2, 5 ng/mL IL-12, and Dynabeads Mouse T-activator CD3/CD28 (Thermo Fisher) (8.3 CD8$^+$ T:Beads = 1:1) in the presence or absence of BMS-986165 at 37 °C and 5% $CO_2$ for 3d. Then, 1 ×10$^5$ NIT-1 cells were cultured in flat-bottom 96-well plates in F-12K with 10% FBS, 1% PcSM, and incubated at 37 °C and 5% $CO_2$ for 24 h. For the co-culture cytotoxicity assay, ex vivo activated 8.3 CD8$^+$ T cells were added to a plate with NIT-1 cells, and co-cultured in F-12K with 10% FBS, 1% PcSM, and 1 µg/mL IGRP$_{206-214}$ (VYLKTNVFL) at 37 °C and 5% $CO_2$ for 24 h (NIT-1:CD8$^+$ T ratio = 1:0.1, 1:1, 1:2). To assess the viability of NIT-1 cells, the supernatants were removed and washed three times with PBS(-) to remove 8.3 CD8$^+$ T cells. After washing, 100 µL of F-12K with 10% FBS, 1% PcSM, and 20 µL of CellTiter 96 (Promega, Tokyo, Japan) were added and incubated at 37 °C and 5% $CO_2$ for 2 h. The absorbance was determined at 450 nm. The % killing was calculated as follows: % killing = 100 − ((Abs of NIT-1 cells cultured with 8.3 CD8$^+$ T cells)/(Abs of untreated NIT-1 cells) × 100). To analyze the gene expression in NIT-1 cells, supernatants were removed and washed three times with PBS(-) to remove 8.3 CD8$^+$ T cells. After washing, RNA was extracted using Isogen II (Nippon gene) and cDNA was synthesized using a High-capacity cDNA reverse transcription kit (Applied Biosystems) according to the manufacturer's instructions.

To assess the cytokine production in 8.3 CD8$^+$ T cells, ex vivo activated 8.3 CD8$^+$ T cells and NIT-1 cells were co-cultured in F-12K with 10% FBS, 1% PcSM, and 1 µg/mL IGRP$_{206-214}$ (VYLKTNVFL) at 37 °C and 5% $CO_2$ for 24 h (NIT-1:CD8$^+$ T ratio = 1:1) with or without 10 µg/mL of antibodies (anti-mouse IFNAR1 antibody (MAR1-5A3; Leinco Technologies, Fenton, MO, USA), mouse IgG1 isotype control (MOPC-21; BioXCell, Lebanon, NH, USA), anti-mouse IFN-β antibody (HDB-4A7; ichorbio, Wantage, UK), mouse IgG2a isotype control (BioXCell), anti-mouse IFN-α antibody (TIF-3C5; ichorbio), and Armenian hamster IgG isotype control (RIP; BioXCell)), and 40 µg/mL Brefeldin A (Sigma-Aldrich) was added for the last 4 h of incubation. To test the Caspase 3 and 7 activation in 8.3 CD8$^+$ T cells, the cells were incubated with CellEvent Caspase-3/7 green flow cytometry assay kit (Invitrogen) according to the manufacture's protocol.

### BMS-986165
BMS-986165 (deucravacitinib) (6-cyclopropaneamido-4-{[2-methoxy-3-(1-methyl-1H-1,2,4-triazol-3-yl)phenyl]amino}-N-(2H3)methylpyr-idazine-3-carboxamide) was purchased from Selleck Biotech (Tokyo, Japan). To test the inhibitory effects of BMS-986165 on the development of autoimmune T1D, wild type NOD mice were dosed by oral gavage once daily with BMS-986165 30 mg/kg in vehicle (EtOH:TPG-S:PEG300, 5:5:90) or vehicle alone as previously described[53].

To assess the inhibitory effects of BMS-986165 in the priming of CTLs ex vivo, CD8$^+$ T cells were isolated using two rounds of the CD8$^+$ T cell isolation kit (Miltenyi Biotec) with LS columns (Miltenyi Biotec) and cultured with 30 U/mL IL-2, 5 ng/mL IL-12, and Dynabeads Mouse T-activator CD3/CD28 (Thermo Fisher) (CD8$^+$ T:beads = 1:1) in the presence or absence of BMS-986165 at 37 °C and 5% $CO_2$ for 3d. To test the induction of CXCR3 in CTLs, primed CTLs were washed and re-cultured in U-bottom 96-well plates in RPMI-1640 with 10% FBS, 1% PcSM, 50 µM 2-mercaptoethanol at 37 °C and 5% $CO_2$ for 2d. To assess the inhibitory effects of BMS-986165 in bystander activation of T cells, 5 ×10$^5$ splenocytes were cultured in 12-well plates in RPMI-1640 with 10% FBS, 1% PcSM, 50 µM 2-mercaptoethanol, and stimulated with 20 ng/mL IL-12 (BioLegend) and 20 ng/mL IL-18 (BioLegend) at 37 °C and 5% $CO_2$ for 24 h with or without BMS-986165; 40 µg/mL Brefeldin A (Sigma-Aldrich) was added for the last 5 h of the incubation. To assess the inhibitory effects of BMS-986165 in cytokine-induced gene expressions in NIT-1 cells, 5 ×10$^5$ NIT-1 cells were cultured in 12-well plates in F-12K with 10% FBS and 1% PcSM, and stimulated with 100 U/mL IFN-β (Pbl Assay Science, Piscataway, NJ, USA) and 0.1 ng/mL IFN-γ (BioLegend) at 37 °C and 5% $CO_2$ for 3 h (for *Cxcl9*,

*Cxcl10*, *Cxcl13*, *Isg15*, and *Oas2*) or 24 h (for *Fas*) in the presence or absence of BMS-986165.

### Data analysis
Heatmaps were generated with the R (3.6.0) package pheatmap, and clustering was performed using the word.D2 method. Enrichment analysis was performed by GeneTrail[70]. PCA analysis was performed using ClustVis[71] and RNAseqChef[72]. Volcano plots were generated using ggVolcanoR[73]. Likelihood ratio tests were performed to determine differentially expressed genes between genotypes using RNAseqChef[72].

### Statistical analysis
Data are expressed as the means ± SEM. Statistical significance between two groups was determined using two-tailed Student's *t* test. For the analysis of diabetes incidence, Kaplan–Meier survival curves were estimated by the log rank test. Statistical significance of insulitis levels were analyzed using the Mann–Whitney *U*-test. To analyze differences among three or more groups, statistical significance was determined using one-way ANOVA followed by post hoc Dunnett's or Tukey's multiple-comparisons tests, and two-way ANOVA with Tukey's post hoc test. $P < 0.05$ was considered statistically significant. Statistical analysis was performed using Prism 9 (9.5.1) (GraphPad Software). Differentially expressed genes between pairwise comparisons were performed using DESeq2 (FDR < 0.1, fold change >1.5) in iDEP.96[74]. For the cluster analysis of IGRP CD8$^+$ T cells in the PLN and pancreas, likelihood ratio tests were used to determine differentially expressed genes between the genotypes (FDR < 0.1, fold change >1.5) using RNAseqChef[72]. No a priori statistical methods were used to determine the sample size. Sample size sufficiency was based on previous experiments from our laboratory and others. The investigators were not blinded to the allocation of mice, samples, and outcome data analyses.

### Reporting summary
Further information on research design is available in the Nature Portfolio Reporting Summary linked to this article.

## Data availability
All data analyzed in this study are included in this article and supplementary information, or have been made available in public repositories. The microarray data generated in this study have been deposited in the Gene Expression Omnibus under accession code GSE235670. The RNA-sequencing data generated in this study have been deposited in the Sequence Read Archive under accession code PRJNA992677. The representative flow cytometry data generated in this study have been deposited in the Mendeley Data under https://doi.org/10.17632/b5bybfddnn.1 [https://data.mendeley.com/datasets/b5bybfddnn/1]. Source data are provided with this paper.

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

## Acknowledgements

This work was supported by a Japan IDDM Network Research Grant (S.N.), Japan Diabetes Foundation and Costco Wholesale Japan Ltd. (K.M.), and the NOVARTIS Foundation (Japan) for the Promotion of Science (K.M.). We sincerely thank Shinichiro Sawa, Kota Yanagitani, Shinichi Koizumi, Naoto Noguchi, Shinya Hatano, and Yumiko Tsugitomi for their counsel on experiments, and Yumiko Kitada, Akiko Takano, Chieko Ogawa, Saori Fuchigami, Rasheda Perveen, Yoshifumi Morita, and Hiroaki Wakimoto for helping to prepare the manuscript. We appreciate technical support from the Analytical Research Center for Experimental Sciences, Saga University. We thank Edanz (https://jp.edanz.com/ac) for editing a draft of this manuscript.

## Author contributions

Conceptualization, K.M., K.A., and S.N.; Investigation, K.M.; writing - original draft preparation, K.M.; writing - review and editing, H.T., H.M., S.A., N.A., H.K., K.S., K.A., Y.Y., and S.N.; supervision, S.N.; project administration, K.M.; resources, S.A., N.A., K.S., Y,Y., and H.K.; funding acquisition, K.M., and S.N. All authors have read and agreed to the published version of the manuscript.

## Competing interests

The authors declare no competing interests.
