## [Peer Review File · Nature Communications]

REVIEWER COMMENTS

Reviewer #1 (expert in type-1 diabetes):

There has been progressive understanding of functions of tyrosine kinase 2 (Tyk2) beyond type 1 interferons. However, the role of Tyk2 in type 1 diabetes (T1D) is less clear. Tyk2 was reported to be associated with acute-onset of T1D, post viral infection; however, the role of Tyk2 in commonly slow-progressing T1D development has not been known. The authors generated Tyk2 deficient NOD mice, the well-received mouse model of human T1D, and performed a thorough investigation of the Tyk2 knockout (KO) NOD mice. Tyk2 deficient NOD mice have delayed and markedly reduced incidence of T1D. To investigate the mechanism underlying T1D protection, the authors took cellular and molecular approaches and performed many experiments. The main findings, among others, were that Tyk2 deficiency led to impaired function of CD8+ T cells and CD8+ DCs, as well as attenuated expression of some molecules associated with immune responses.

This is an interesting study which provides novel knowledge about the role of Tyk2 in T1D development. However, this reviewer has the following concerns and comments:

Major:

To investigate the impaired functions of CD8+ T cells in the absence of Tyk2, the authors generated Tyk2 deficient NY8.3 CD8 TCR transgenic NOD mice. The authors used Tyk2 deficient NY8.3 CD8+ T cells for some in vitro cytotoxicity and in vivo proliferation assays. However, it is important to know if diabetes development would be delayed and/or reduced in Tyk2 deficient NY8.3 NOD mice, especially as the authors found impaired function(s) of CD8+ T cells in the absence of Tyk2. The Tyk2 deficient NY8.3 NOD mouse is the perfect tool to confirm and/or validate the authors' finding and the authors have these mice in hand.

Minor but not trivial:

1). T1D genetic susceptibility loci are more than those the author have detected using PCR. Although this reviewer thinks that the authors' Tyk2^{-/-} NOD mice are likely to be mostly on NOD genetic background, a more rigorous test would be to carry out the genome scan by SNP linkage mapping. This can be done easily by various service providers, one of which is DartMouseTM (<https://geiselmed.dartmouth.edu/dartmouse/>).

2). Fig.1B. What does the insulinitis look

like in 24-week-old +/+ or +/- mice? The +/+ and +/- mice showed that about 35% and 60% mice respectively, had become diabetic by this time. Thus, there were still 65% and 40% of mice, respectively, which had not become diabetic at 24-weeks-old.

3). Fig.1C. What about the IAA in 24-week-old +/+ and +/- mice? Without the data from +/+ or +/- mice at a similar age, it is difficult to interpret the results that showed that IAA was significantly increased in -/- mice. The authors' interpretation related to aging is somewhat simplistic.

4). On line 194-195 of the main text, the authors stated that MHC class I, H-2K, was reduced in expression on islet beta cells whereas B2m was more highly expressed. The authors did not show B2m

expression. Further, although H2-K appears to be the more highly expressed MHC class I molecule in NOD mice, it would be interesting to know about the expression of H-2D, the 2nd major MHC class I molecule in NOD mice. This also applies to Fig.S4 F and G.

5). In the context of the lower expression of Fas and H2-K, what happened to beta cell function, such as insulin production? Was this also changed in the absence of Tyk2? This question could shed light on whether beta cells are more self-protected by down regulating molecules related to autoimmune attack or the beta cells may also down regulate production of self antigen(s).

6). Fig.4A the authors showed +/- 24w/-, what about -/- 24w+/-? The authors also showed +/- 10 days data, what about -/- +/- 10 day results? This also applies to Fig.S4B.

7) Fig.4B, the authors showed that the proportion of IGRP tetramer positive CD8+ T cells remained stable between 6-wk-old and 11-wk-old mice. Based on the data, the author drew the conclusion that there was no epitope spreading. This reviewer does not think these data support the conclusion, as epitope spreading does not mean the alteration of a tested and disease-causing epitope.

8). The authors showed CD44hi T cells, in most if not all, data, and the status of CD62L of the CD44hi T cells was not clear. It is also not clearly stated why the authors focused on CD44hi T cell populations.

9). What was the gating used in Fig.5E?

10). Based on some reduction of CXCR3+ CD8+ T cells in PLN but not iLN and spleen (Fig.5F, G), the authors thought those T cells exited PLN and migrated to islets. This would be supported if the authors showed the islet CD8+ T cells are CXCR3+.

11). It would be important to test cross priming in the absence of IGRP peptide as the assay will test direct antigen presentation in the presence of peptide, but not cross presentation. The authors should test NY8.3 T cells in response to NIT-1 cells in the presence of DCs but in absence of IGRP peptide. Alternatively, the authors need to pulse DCs with NIT-1 cells and use the pulsed DCs as antigen presenting cells to elicit NY8.3 T cell responses.

12). The authors used 10 µg/ml IGRP peptide in their assays; this concentration of peptide is extremely high for NY8.3 CD8+ T cells.

13). The authors showed that Tyk2-/- IGRP CD8+ T cells had upregulated apoptosis-related genes including Casp3 and Birc5 (Fig.5I), it is not clear if those cells are indeed prone to apoptosis. Annexin V staining would provide some information and/or explanation as to why the cells upregulated apoptosis-related genes.

14). The earliest time when the wild type (or +/-) NOD mice developed T1D was around 14 to 15-wks-old (Fig.1A), whereas the earliest age when Tyk2-/- NOD mice developed T1D was ~22-wk-old (Fig.1A). Early (from 6-wk-old) treatment of Tyk2 inhibitor BMS-986165 in wild type NOD mice showed further delay in T1D onset, ~32-wk-old, and marked reduced T1D incidence, ~20% (Fig.6F), which is similar to the Tyk2-/- mice shown in Fig.1A. Tyk2 inhibitor treatment at pre-diabetic period (from 12 -wk-old) also strikingly delayed T1D onset (the first mouse developed diabetes was at 30-wk-old, Fig.6G). This reviewer assumes that if the group size were larger, especially the treatment group, which had only 4 mice, the statistical outcome could be significant, even though the overall incidence of diabetes was not reduced.

15). Many figure legends did not provide brief description about how the experiments were conducted. In this reviewer's opinion, the figure legend should be more informative, and the readers should not need to go to and from the main text and/or Materials/Methods to find the information about the figures. Further, it is not clear why the authors used MFI as the readout for most, if not all, the flow cytometric analysis.

Reviewer #2 (expert in JAK-STAT signalling and TYK2):

The manuscript adds important insight and novelty in the cellular mechanisms in TYK2-mediated autoimmune type 1 diabetes and its potential prevention by a TYK2inib in preclinical mouse models.

Overall, the impressive data sets support the conclusions and are appropriately presented. Previously published, mostly the author's claims supporting work should be included and discussed appropriately (see comments)

The methodology is state-of-the-art, provided details allow for reproducibility (but see comments)

Major comments:

- The authors should include in their introduction/discussion the relevant findings of Chandra et al (<https://doi.org/10.1038/s41467-022-34069-z>) in a human TYK2-deficient cell system
- Fig2C the authors should consider to complement the heat map by an alternative visualisation of the crucial DEG and indicate in the results text more clearly which DEG (esp. IFNalpha and IFNbeta) might be of biological relevance albeit not reaching the applied statistical significance criteria. In this context the authors should also mention previous supportive work on human pancreatic beta-cells producing IFNalpha and CXCL10 under challenge conditions in a TYK2-dependent manner (Marroqui et al, <https://doi.org/10.2337/db15-0362>)
- line 337ff, rationale for next experiments: the authors should acknowledge that impaired in vitro and in vivo killing capacity of TYK2-deficient CTLs has been already described (e.g. Simma et al, <https://doi.org/10.1158/0008-5472.CAN-08-1705>).
- line 477ff, discussion on potential risks/side effects of TYK2inib treatment: the authors should also mention the involvement of TYK2 in tumour surveillance and the potential cancer risk upon long-term deucravacitinib exposure (e.g. Yarmolinsky et al., <https://doi.org/10.1002/ijc.34180>).
- Technical/methodological details necessary for the understanding of the presented data a scattered rather randomly in the manuscript; e.g. insulinitis scores 0-3 are defined in the legend to Figure S1 and not given in Material and Methods 'Histology'; statistical stringency for RNASeq DEG is given in Material and Methods only and would help the reader if also mentioned in the results text or given in the legends of the main figures.

The authors are asked to revised the entire manuscript and indicate the necessary technical details in a clear and coherent manner.

Minor comments:

Line 129-30: The journal strongly encourages researchers to follow the 'Sex and Gender Equity in Research – SAGER – guidelines' and to include sex and gender considerations for studies involving humans, vertebrate animals and cell lines where relevant to the topic of study (an overview can be found here). The authors should comment on their findings.

Figure 4E: the number of labelled DEG confuses the take home message of the figure part, the authors could remove labels of non-ISGs and DEGs irrelevant of DC function

RESPONSES TO REVIEWER 1:

Dear Reviewer #1 (expert in type 1 diabetes),

We wish to thank the Reviewer for your insightful comments, which have greatly helped us to improve the quality of our paper. In particular, the amount of IGRP peptide is critical for our experiments. Thank you very much for pointing this out.

The Reviewer's comments are written in blue. Our point-by-point responses to the Reviewer's comments are shown below.

Major: To investigate the impaired functions of CD8⁺ T cells in the absence of Tyk2, the authors generated Tyk2 deficient NY8.3 CD8 TCR transgenic NOD mice. The authors used Tyk2 deficient NY8.3 CD8⁺ T cells for some in vitro cytotoxicity and in vivo proliferation assays. However, it is important to know if diabetes development would be delayed and/or reduced in Tyk2 deficient NY8.3 NOD mice, especially as the authors found impaired function(s) of CD8⁺ T cells in the absence of Tyk2. The Tyk2 deficient NY8.3 NOD mouse is the perfect tool to confirm and/or validate the authors' finding and the authors have these mice in hand.

Response: We thank the Reviewer for this comment. We have provided the data of the incidence of spontaneous diabetes in female NY8.3 NOD mice in Supplementary Fig. 4C (shown below) (*Tyk2*^{+/+} n=8, *Tyk2*^{+/-} n=7, and *Tyk2*^{-/-} n=10). The diabetes incidence was slightly reduced in *Tyk2*^{-/-} NY8.3 NOD mice (60%) compared with *Tyk2*-sufficient NY8.3 NOD mice (*Tyk2*^{+/+}, 87.5%; *Tyk2*^{+/-}, 85.7%) and the timing of diabetes onset was comparable between the genotypes.

We have added the following text in the Results and discussed these results.

(Line 285-292) "NY8.3-NOD transgenic mice develop diabetes much earlier than wild type NOD mice⁴¹. To assess the role of TYK2 under highly diabetogenic conditions, we analyzed the incidence of diabetes and found that it was slightly reduced in *Tyk2*^{-/-} NY8.3 NOD mice (60%) compared with *Tyk2*-sufficient NY8.3 NOD mice (*Tyk2*^{+/+}, 87.5%; *Tyk2*^{+/-}, 85.7%) (Supplementary Fig. 4C). The timing of diabetes onset was comparable between the genotypes (Supplementary Fig. 4C). These results suggest that *Tyk2* deficiency is partially involved in the suppression of diabetes development under artificial conditions of abundant diabetogenic CD8 T cells."

Revised Supplementary Fig. 4C. The incidence of diabetes in female NY8.3 NOD mice ($Tyk2^{+/+}$ n=8, $Tyk2^{+/-}$ n=7, and $Tyk2^{-/-}$ n=10). *P*-values were calculated using the log-rank test.

Minor but not trivial:

1). T1D genetic susceptibility loci are more than those the author have detected using PCR. Although this reviewer thinks that the authors' $Tyk2^{-/-}$ NOD mice are likely to be mostly on NOD genetic background, a more rigorous test would be to carry out the genome scan by SNP linkage mapping. This can be done easily by various service providers, one of which is DartMouse™ (<https://geiselmed.dartmouth.edu/dartmouse/>).

Response: The Reviewer's comment is correct. We mistakenly described the results of the PCR analysis of T1D susceptibility loci as "all T1D genetic susceptibility loci...". This error has been corrected in accordance with the comment (as shown below).

We tested 64 short tandem repeats in two $Tyk2^{-/-}$ NOD mice using the service provider (ICLAS Monitoring Center, Japan. <https://www.iclasmonic.jp/>). The results are shown in Supplementary Fig. 1A and below.

We have revised the Results as described below.

(Line 125-129) "We analyzed 64 short tandem repeat (STR) loci in $Tyk2$ KO.NOD mice and confirmed that all tested loci except for D9Mit83, which is located in chromosome 9 that has $Tyk2$ gene, were of NOD origin (Supplementary Fig. 1A). In addition, we confirmed STRs at the insulin-dependent diabetes susceptibility (Idd) loci (Idd1 to Idd15) were of NOD origin (Supplementary Fig. 2B)²⁸."

Marker	Chr.	Position (cM)	C57BL/6Jel		NOD/Shi		CG1197#01		CG1197#02	
			1	2	1	2	1	2	1	2
D1Mit67	1	4.12	B6	B6	NOD	NOD	NOD	NOD	NOD	NOD
D1Mit132	1	39.51	B6	B6	NOD	NOD	NOD	NOD	NOD	NOD
D1Mit102	1	63.32	B6	B6	NOD	NOD	NOD	NOD	NOD	NOD
D1Mit459	1	91.86	B6	B6	NOD	NOD	NOD	NOD	NOD	NOD
D2Mit1	2	2.23	B6	B6	NOD	NOD	NOD	NOD	NOD	NOD
D2Mit182	2	39.53	B6	B6	NOD	NOD	NOD	NOD	NOD	NOD
D2Mit311	2	86.02	B6	B6	NOD	NOD	NOD	NOD	NOD	NOD
D2Mit346	2	100.84	B6	B6	NOD	NOD	NOD	NOD	NOD	NOD
D3Mit149	3	1.96	B6	B6	NOD	NOD	NOD	NOD	NOD	NOD
D3Mit25	3	27.32	B6	B6	NOD	NOD	NOD	NOD	NOD	NOD
D3Mit85	3	71.03	B6	B6	NOD	NOD	NOD	NOD	NOD	NOD
D3Mit89	3	80.79	B6	B6	NOD	NOD	NOD	NOD	NOD	NOD
D4Mit227	4	4.43	B6	B6	NOD	NOD	NOD	NOD	NOD	NOD
D4Mit52	4	52.62	B6	B6	NOD	NOD	NOD	NOD	NOD	NOD
D4Mit256	4	86.16	B6	B6	NOD	NOD	NOD	NOD	NOD	NOD
D5Mit146	5	3.43	B6	B6	NOD	NOD	NOD	NOD	NOD	NOD
D5Mit58	5	38.44	B6	B6	NOD	NOD	NOD	NOD	NOD	NOD
D5Mit367	5	60.43	B6	B6	NOD	NOD	NOD	NOD	NOD	NOD
D5Mit97	5	76.10	B6	B6	NOD	NOD	NOD	NOD	NOD	NOD
D6Mit86	6	1.81	B6	B6	NOD	NOD	NOD	NOD	NOD	NOD
D6Mit284	6	41.11	B6	B6	NOD	NOD	NOD	NOD	NOD	NOD
D6Mit304	6	78.22	B6	B6	NOD	NOD	NOD	NOD	NOD	NOD
D7Mit267	7	17.09	B6	B6	NOD	NOD	NOD	NOD	NOD	NOD
D7Mit350	7	47.43	B6	B6	NOD	NOD	NOD	NOD	NOD	NOD
D7Mit105	7	70.29	B6	B6	NOD	NOD	NOD	NOD	NOD	NOD
D8Mit155	8	2.14	B6	B6	NOD	NOD	NOD	NOD	NOD	NOD
D8Mit191	8	21.16	B6	B6	NOD	NOD	NOD	NOD	NOD	NOD
D8Mit88	8	61.66	B6	B6	NOD	NOD	NOD	NOD	NOD	NOD
D9Mit83	9	8.39	B6	B6	NOD	NOD	B6	B6	B6	B6
D9Mit97	9	27.75	B6	B6	NOD	NOD	NOD	NOD	NOD	NOD
D9Mit52	9	73.06	B6	B6	NOD	NOD	NOD	NOD	NOD	NOD
D10Mit2	10	9.78	B6	B6	NOD	NOD	NOD	NOD	NOD	NOD
D10Mit31	10	35.26	B6	B6	NOD	NOD	NOD	NOD	NOD	NOD
D10Mit266	10	63.23	B6	B6	NOD	NOD	NOD	NOD	NOD	NOD
D11Mit21	11	25.94	B6	B6	NOD	NOD	NOD	NOD	NOD	NOD
D11Mit67	11	60.47	B6	B6	NOD	NOD	NOD	NOD	NOD	NOD
D11Mit48	11	82.96	B6	B6	NOD	NOD	NOD	NOD	NOD	NOD
D12Mit109	12	18.94	B6	B6	NOD	NOD	NOD	NOD	NOD	NOD
D12Mit156	12	35.70	B6	B6	NOD	NOD	NOD	NOD	NOD	NOD
D12Mit133	12	60.56	B6	B6	NOD	NOD	NOD	NOD	NOD	NOD
D13Mit57	13	5.92	B6	B6	NOD	NOD	NOD	NOD	NOD	NOD
D13Mit13	13	30.06	B6	B6	NOD	NOD	NOD	NOD	NOD	NOD
D13Mit51	13	56.45	B6	B6	NOD	NOD	NOD	NOD	NOD	NOD
D14Mit1	14	6.33	B6	B6	NOD	NOD	NOD	NOD	NOD	NOD
D14Mit233	14	26.83	B6	B6	NOD	NOD	NOD	NOD	NOD	NOD
D14Mit225	14	39.46	B6	B6	NOD	NOD	NOD	NOD	NOD	NOD
D15Mit13	15	1.84	B6	B6	NOD	NOD	NOD	NOD	NOD	NOD
D15Mit171	15	45.02	B6	B6	NOD	NOD	NOD	NOD	NOD	NOD
D15Mit42	15	55.72	B6	B6	NOD	NOD	NOD	NOD	NOD	NOD
D16Mit129	16	2.71	B6	B6	NOD	NOD	NOD	NOD	NOD	NOD
D16Mit139	16	37.28	B6	B6	NOD	NOD	NOD	NOD	NOD	NOD
D16Mit106	16	57.68	B6	B6	NOD	NOD	NOD	NOD	NOD	NOD
D17Mit223	17	2.66	B6	B6	NOD	NOD	NOD	NOD	NOD	NOD
D17Mit53	17	38.64	B6	B6	NOD	NOD	NOD	NOD	NOD	NOD
D17Mit93	17	45.20	B6	B6	NOD	NOD	NOD	NOD	NOD	NOD
D18Mit12	18	19.29	B6	B6	NOD	NOD	NOD	NOD	NOD	NOD
D18Mit91	18	29.67	B6	B6	NOD	NOD	NOD	NOD	NOD	NOD
D18Mit187	18	50.99	B6	B6	NOD	NOD	NOD	NOD	NOD	NOD
D19Mit78	19	5.33	B6	B6	NOD	NOD	NOD	NOD	NOD	NOD
D19Mit14	19	14.32	B6	B6	NOD	NOD	NOD	NOD	NOD	NOD
D19Mit103	19	48.46	B6	B6	NOD	NOD	NOD	NOD	NOD	NOD
DXMit55	X	3.31	B6	B6	NOD	NOD	NOD	NOD	NOD	NOD
DXMit25	X	36.78	B6	B6	NOD	NOD	NOD	NOD	NOD	NOD
DXMit121	X	73.95	B6	B6	NOD	NOD	NOD	NOD	NOD	NOD
Marker	Chr.	Position (cM)	C57BL/6Jel		NOD/Shi		CG1197#01		CG1197#02	
			1	2	1	2	1	2	1	2

Revised Supplementary Fig. 1A. The analysis of 64 short tandem repeat (STR) loci in female *Tyk2*KO.NOD mice.

2). Fig.1B. What does the insulinitis look like in 24-week-old $+/+$ or $+/-$ mice? The $+/+$ and $+/-$ mice showed that about 35% and 60% mice respectively, had become diabetic by this time. Thus, there were still 65% and 40% of mice, respectively, which had not become diabetic at 24-weeks-old.

3). Fig.1C. What about the IAA in 24-week-old $+/+$ and $+/-$ mice? Without the data from $+/+$ or $+/-$ mice at a similar age, it is difficult to interpret the results that showed that IAA was significantly increased in $-/-$ mice. The authors' interpretation related to aging is somewhat simplistic.

Response to 2 and 3: In accordance with the Reviewer's comment, we analyzed the insulinitis scores of normoglycemic 24-week-old $Tyk2^{+/+}$ (n=10) and $Tyk2^{+/-}$ NOD mice (n=8). Percentage of islets with insulinitis in 24-week-old $Tyk2^{+/+}$ and $Tyk2^{+/-}$ mice was comparable to that in 14-week-old mice. Because diabetic mice were excluded, these results understates the percentage of islets with insulinitis, especially in 24-week-old $Tyk2^{+/+}$ and $Tyk2^{+/-}$ mice.

Revised Fig. 1B. Percentage of islets with a given insulinitis score.

P-values were calculated using the one-way ANOVA with Tukey's posttest.

We analyzed the levels of serum IAAs of normoglycemic 24-week-old $Tyk2^{+/+}$ (n=10) and $Tyk2^{+/-}$ NOD mice (n=8). Levels of IAAs were comparable among 24-week-old mice irrespective of genotypes. This result also understates levels of IAA in 24-week-old mice.

Revised Fig. 1C. Levels of serum IAAs.

P-values were calculated using the one-way ANOVA with Tukey's posttest.

On the basis of these results, we have changed the following text from:

“We also analyzed 24w mice to assess the long-term effects of *Tyk2* deficiency. Increased numbers of inflamed islets and elevated levels of serum IAAs were observed in 24w *Tyk2*^{-/-} mice compared with 14w *Tyk2*^{-/-} mice (Fig. 1B, C). These observations suggest that *Tyk2* deficiency does not completely prevent islet autoimmunity but reduces the progression rate of invasive insulinitis leading to T1D onset.”

to

(line 144-151) “We also analyzed normoglycemic 24w mice to assess the long-term effects of *Tyk2* deficiency. Increased numbers of inflamed islets and elevated levels of serum IAAs were observed in 24w *Tyk2*^{-/-} mice compared with 14w *Tyk2*^{-/-} mice (Fig. 1B, C). Because diabetic mice were excluded, levels of insulinitis and IAAs were understated in this analysis, especially in 24w *Tyk2*^{+/+} and *Tyk2*^{+/-} mice. These observations suggest that *Tyk2* deficiency does not completely prevent islet autoimmunity but reduces the progression rate of invasive insulinitis leading to T1D onset.”

4). On line 194-195 of the main text, the authors stated that MHC class I, H-2K, was reduced in expression on islet beta cells whereas B2m was more highly expressed. The authors did not show B2m expression. Further, although H2-K appears to be the more highly expressed MHC class I molecule in NOD mice, it would be interesting to know about the expression of H-2D, the 2nd major MHC class I molecule in NOD mice. This also applies to Fig.S4 F and G.

Response: We appreciate the Reviewer's comment, which adds further insights to our study. In accordance with Reviewer 2's comments, we have revised Fig. 2. The heatmap figure was changed to an alternative visualization using Z-scores, which was also used to create the previous heatmap. In revised Fig. 2D, normalized expression levels of *B2m* and *H-2k1* are shown. Because the microarray does not have a probe for *H-2d*, the gene expression data of *H-2d* in β-cells are not shown in Fig. 2D.

Revised Fig. 5D,E (Response to Reviewer 2's comments). Expression levels of selected genes in β -cells. Data are normalized by Z score. Asterisks (*) indicate differentially expressed genes between 11w WT and 11w KO mice. Daggers (†) indicate differentially expressed genes between 6w WT vs 11w WT mice. Double daggers (‡) indicate differentially expressed genes between 6w KO and 11w KO mice.

Instead, we have analyzed the protein expression levels of H-2D^b in β -cells (Fig. 2F (right), shown below). The expression levels of H-2D^b in β -cells were comparable between *Tyk2* genotypes.

On the basis of these results, we have changed the following text from:

“The protein expression levels of MHC I in β -cells were slightly decreased in 11–12w *Tyk2*^{-/-} mice compared with age-matched littermates (Figure 2D).”

to

(Line 206-209) “We also analyzed the protein expression levels of MHC I in β -cells and revealed that H-2K^d, but not H-2D^b, was slightly decreased in 11–12w *Tyk2*^{-/-} mice compared with age-matched littermates (Fig. 2F).”

Revised Fig. 2F. Expression levels of H-2K^d and H-2D^b in β -cells. Purified β -cells from 11 to 12w mice were analyzed using flow cytometry (H-2K^d: *Tyk2*^{+/+}, n=9; *Tyk2*^{+/-}, n=12; *Tyk2*^{-/-}, n=9) (H-2D^b: *Tyk2*^{+/+}, n=7; *Tyk2*^{+/-}, n=9; *Tyk2*^{-/-}, n=9). *P*-values were calculated using Kruskal-Wallis test with Dunn's posttest.

We have also analyzed the protein expression levels of H-2D^b in DCs (Fig. 4D and Supplementary Fig. 5E, G). We found that *Tyk2* deficiency reduced the expression levels of H-2D^b in CD8⁺ rDC but not other subsets of DCs in the PLN (Fig. 4D (right) and Supplementary Fig. 5E). In the iLN and spleen, reduced expression levels of H-2D^b in CD8⁺ rDC were also observed (Supplementary Fig. 5G). Thus, the expression levels of H-2D^b in CD8⁺ rDC were decreased in *Tyk2*^{-/-} NOD mice.

On the basis of these results, we have changed the following text from:

“However, we found that *Tyk2* deficiency reduced the expression of MHC I (H2-K^d), a crucial molecule for presenting antigens to CD8 T cells ³⁸, in CD8⁺ rDC but not other subsets of DCs (Figures 4D and S4F).”

to

(line 306-309) “However, we found that *Tyk2* deficiency reduced the expression of MHC I (H-2K^d and H-2D^b), a crucial molecule for presenting antigens to CD8 T cells ⁴⁰, in CD8⁺ rDC but not other subsets of DCs (Fig. 4D and Supplementary Fig. 5E, G).”

Revised Fig. 4D. Expression levels of H-2K^d and H-2D^b in CD8⁺ rDC in the PLN. *P*-values were calculated using Kruskal-Wallis test with Dunn's posttest.

Revised Supplementary Fig. 5E. Expression levels of H-2D^b in DCs in the PLN. *P*-values were calculated using Kruskal-Wallis test with Dunn's posttest.

Revised Supplementary Fig. 5G. Expression levels of H-2D^b in CD8⁺ rDC in the iLN and spleen. *P*-values were calculated using Kruskal-Wallis test with Dunn's posttest.

5). In the context of the lower expression of Fas and H2-K, what happened to beta cell function, such as insulin production? Was this also changed in the absence of Tyk2? This question could shed light on whether beta cells are more self-protected by down regulating molecules related to autoimmune attack or the beta cells may also down regulate production of self antigen(s).

Response: In accordance with the Reviewer's comment, we checked the gene expression data (microarray) of T1D-related autoantigens in β -cells and found that *Ins1* and IGRP (*G6pc2*) were expressed at lower levels in β -cells from *Tyk2*^{-/-} mice compared with those from *Tyk2*^{+/+} mice at 6 weeks of age (Supplementary Fig. 1G). At 11 weeks of age, gene expression levels of T1D-related autoantigens including *Ins1*, *Ins2*, GAD65 (*Gad2*), IA-2 (*Ptprn*), ZnT-8 (*Slc30a8*), and *G6pc2* in β -cells were comparable between genotypes.

On the basis of these results, we have added the following text to the Results:

(Line 212-218) "Type I IFN signaling in β -cells was associated with the presentation of autoantigens^{19,35}; therefore, we compared the expression levels of T1D-related autoantigens in β -cells. At 11w, gene expression levels of T1D-related autoantigens including insulin (*Ins1* and *Ins2*), GAD65 (*Gad2*), IA-2 (*Ptprn*), ZnT-8 (*Slc30a8*), and IGRP (*G6pc2*) in β -cells were comparable between genotypes (Supplementary Fig.1G). However, at 6w, *Ins1* and *G6pc2* were expressed at lower levels in β -cells from *Tyk2*^{-/-} mice compared with *Tyk2*^{+/+} mice (Supplementary Fig. 1G)."

We have also added the following text to the Discussion:

(Line 503-509) "In the early phase of T1D development, TYK2 inhibition might reduce the expression levels of islet-associated autoantigens in β -cells (Supplementary Fig. 1G). In contrast to these preservation effects of TYK2 inhibition in β -cells, a recent study suggested that TYK2 inhibition lead to

reduced β -cell mass related to the defective formation of pancreatic endocrine precursors³⁵. Consequently, TYK2 may have a different role in β -cells depending on the stage of T1D and/or developmental stage of β -cells.”

Revised Supplementary Fig. 1G. Gene expression levels of islet-associated autoantigens in β -cells. Section signs (§) indicate differentially expressed genes between 6w *Tyk2* wild type and 6w *Tyk2* KO β -cells.

6). Fig.4A the authors showed +/- \diamond 24w-/-, what about -/- \diamond 24w+/-? The authors also showed +/- \diamond -/- 10 days data, what about -/- \diamond +/- 10 day results? This also applies to Fig.S4B.

Response: In accordance with the Reviewer’s comment, we performed the requested transfer experiments. The *Tyk2*^{-/-} naïve 8.3 CD8 T cells underwent increased proliferation in the PLN of 24w *Tyk2*^{+/-} recipient mice 5d post transfer compared with those in 24w *Tyk2*^{-/-} recipient mice (Fig. 4A). Ten days after transfer, the *Tyk2*^{-/-} naïve 8.3 CD8 T cells underwent increased proliferation in the PLN of 6 to 8w *Tyk2*^{+/-} recipient mice compared with those 5d after transfer (Fig. 4A).

Revised Fig. 4A. Proliferation of naïve 8.3 CD8 T cells in the PLN: 4×10^6 CTV labeled naïve 8.3 CD8 T cells were transferred into 6-9w or 24w normoglycemic recipient mice.

In the iLN, the *Tyk2*^{-/-} naïve 8.3 CD8 T cells underwent comparable proliferation in 24w *Tyk2*^{+/-} recipient mice 5d post transfer compared with those in 24w *Tyk2*^{-/-} recipient mice (Fig. 4A). Ten days after transfer, the *Tyk2*^{-/-} naïve 8.3 CD8 T cells underwent increased proliferation in the iLN of 6 to 8w *Tyk2*^{+/-} recipient mice compared with those 5d after transfer (Fig. 4A).

Revised Supplementary Fig. 4B. Proliferation of naïve 8.3 CD8 T cells in the iLN: 4×10^6 CTV labeled naïve 8.3 CD8 T cells were transferred into 6-9w or 24w recipient mice.

On the basis of these results, we have changed the following text from:

“Ten days after the transfer, *Tyk2*^{+/-} naïve 8.3 CD8 T cells showed greater proliferation in the PLN of *Tyk2*^{-/-} recipient mice compared with 5 days post transfer (Figure 4A). The limited proliferation of 8.3 CD8 T cells was observed in the iLN (Figure S4B), indicating the importance of the PLN for the proliferation of islet-autoreactive CTLs. Together, these observations suggest that CD8 T cell-extrinsic mechanisms have a role in the proliferation of islet-autoreactive CTLs in the PLN.”

to

(Line 264-273) “These observations suggest that CD8 T cell-extrinsic mechanisms have a role in the proliferation of islet-autoreactive CTLs in the PLN. Indeed, *Tyk2*^{-/-} 8.3 CD8 T cells more proliferated in the PLN of *Tyk2*^{+/-} recipient mice 10 days after transfer compared with those 5 days post transfer (Fig. 4A). Ten days after transfer, the number of proliferating 8.3 CD8 T cells was increased in the PLN of *Tyk2*^{-/-} mice compared with those 5 days after transfer (Fig. 4A), suggesting that the priming of CD8 T cells was impaired but not abolished in the PLN of *Tyk2*^{-/-} mice. In agreement with a study showing the importance of the PLN for the proliferation of islet-autoreactive CTLs³⁹, the limited proliferation of 8.3 CD8 T cells was observed in the iLN (Supplementary Fig. 4B).”

7) Fig.4B, the authors showed that the proportion of IGRP tetramer positive CD8+ T cells remained stable between 6-wk-old and 11-wk-old mice. Based on the data, the author drew the conclusion that

there was no epitope spreading. This reviewer does not think these data support the conclusion, as epitope spreading does not mean the alteration of a tested and disease-causing epitope.

Response: We think that the Reviewer might be mistaken on this point. We described that “ (line 275-276) the defective proliferation of CD8 T cells in the PLN of *Tyk2*^{-/-} mice is not due to reduced IGRP epitope spreading”.

On the basis of the results that the proportion of IGRP tetramer-positive CD8 T cells remained stable between genotypes (Fig. 4B), we considered that IGRP epitope spreading occurred in the *Tyk2*^{-/-} NOD mice. If there was no IGRP epitope spreading, IGRP-specific CD8 T cells would not be detected in the PLN because the IGRP epitope is not available.

We have revised the following text in the Results to more clarify our findings.

(Line 274-284) “Because IGRP epitopes were reported to appear and spread in the middle-to-late stage of T1D⁴², defects in IGRP epitope spreading in *Tyk2*^{-/-} mice might correlate with the reduced proliferation of 8.3 CD8 T cells in the PLN of *Tyk2*^{-/-} mice. To exclude this possibility, we evaluated IGRP-specific CD8 T cells (IGRP CD8 T) using tetramers containing a mimotope of IGRP (NRP-V7). A comparable frequency of IGRP CD8 T cells was observed in the PLN of both *Tyk2* genotypes (Fig. 4B). Furthermore, the proliferation of 8.3 CD8 T cells was also suppressed in the PLN of normoglycemic 24w *Tyk2*^{-/-} recipient mice with advanced insulinitis compared with 14w *Tyk2*^{-/-} mice, but not in the PLN of normoglycemic 24w *Tyk2*^{+/-} recipient mice (Fig. 1B and 4A). These data suggest that the defective proliferation of CD8 T cells in the PLN of *Tyk2*^{-/-} mice is not due to reduced IGRP epitope spreading in *Tyk2*^{-/-} mice, but other factor(s).”

8). The authors showed CD44^{hi} T cells, in most if not all, data, and the status of CD62L of the CD44^{hi} T cells was not clear. It is also not clearly stated why the authors focused on CD44^{hi} T cell populations.

Response: In accordance with the Reviewer’s comment, we revised texts in the Results to clearly state why we focused on CD44^{hi} T cells as described below:

(Line 255-256) “Antigen encountered naïve T cells become activated, proliferate, and develop into effector and memory T cells that exhibit the CD44^{hi} phenotype⁴⁰.”

(Line 336-338) “Next, we characterized antigen-experienced CD8 T cells that exhibited the CD44^{hi} phenotype. Because transcription factors (TFs) are key regulators of T cell effector functions, we examined TFs in CD44^{hi} CD8 T cells in the PLN.”

We showed the CD44 and CD62L expression status in polyclonal CD8 T cells in the PLN (Fig. 3B and Supplementary Fig. 3C) and pancreas (Supplementary Fig. 3A). However, we did not show these

expression status in IGRP-specific CD8 T cells. In the pancreas, CD44^{hi} IGRP CD8 T cells consisted of a CD62L(-) population only (Supplementary Fig. 6E). In the PLN, CD44^{hi} IGRP CD8 T cells consisted of CD62L(-) and CD62L(+) populations at a ratio of approximately 6:1 (Supplementary Fig. 6E).

We have added the following text.

(Line 383-386): “The expression status of CD44 and CD62L in the IGRP CD8 T cells in the PLN resembled those of polyclonal CD8 T cells (Supplementary Fig. 6E). In the pancreas, IGRP CD8 T cells virtually exhibited a CD62L⁻ phenotype (Supplementary Fig. 6E).”

Revised Supplementary Fig. 6E. Expression status of CD44 and CD62L in IGRP CD8 T cells.

9). What was the gating used in Fig.5E?

Response: The gating strategy in Fig. 5E is shown below (Response Fig. 1). In this experiment, MACS purified CD8 T cells were stimulated *ex vivo* with anti-CD3/CD28 beads, IL-2, and IL-12 for 3d. The T-BET expression levels in CD3⁺CD8⁺ T cells were analyzed (Fig. 5E).

Response Fig. 1. Gating strategy used in Fig. 5E.

10). Based on some reduction of CXCR3⁺ CD8⁺ T cells in PLN but not iLN and spleen (Fig.5F, G), the authors thought those T cells exited PLN and migrated to islets. This would be supported if the authors showed the islet CD8⁺ T cells are CXCR3⁺.

Response: We think that the Reviewer might be mistaken on this admittedly difficult point. In the PLN of *Tyk2*-sufficient mice, more efficient development and proliferation of T-BET⁺ CTLs occurred than in the PLN of *Tyk2* KO mice, whereas the number of CD8 T cells in the PLN was comparable between *Tyk2* genotypes. One reason for this is that the number of naïve CD8 T cells was decreased in the PLN of *Tyk2*-sufficient mice compared with *Tyk2* KO mice. In addition, we considered that the developed CXCR3⁺ CTLs exited the PLN and migrated to the islets where they expressed CXCL10 and CXCL9. It is suggested that the kinetics balance of the development of CTLs and the migration (chemotaxis) of CTLs led to a comparable number of CD8 T cells in the PLN between *Tyk2* genotypes. We thought that a reduction of the number of CXCR3⁺ CD8 T cells in the PLN of *Tyk2* KO mice led to a reduction in the number of CTLs migrating to the pancreas. This putative pathway is shown below (graphical figure).

In accordance with the Reviewer's comment, we have analyzed the expression levels of CXCR3 in CD8 T cells in the pancreas. As shown below, CD8 T cells in the pancreas were not stained by an anti-CXCR3 antibody (Response Fig. 2). There are two possible reasons for this.

Response Fig. 2. Expression of CXCR3 in CD8 T cells in the PLN and pancreas.

One reason is that CXCR3 is known to be downregulated depending on the concentration of CXCL10 or IFN- β (*Journal of Immunology* 2008 (<https://doi.org/10.4049/jimmunol.180.10.6713>), *Cell Reports* 2019 (<https://doi.org/10.1016/j.celrep.2019.06.021>)). Our transcriptome data of β -cells revealed

that β -cells expressed CXCL10 and IFN- β as mice aged (Fig. 2D, E). CXCR3 may be downregulated in pancreatic CD8 T cells due to the presence of high levels of CXCL10 and IFN- β in the pancreas.

The second reason might be technical difficulties. To analyze immune cells in the pancreas, the pancreas was perfused with 1 mg/mL collagenase D, 0.5 mg/mL trypsin inhibitor, and 25 U/mL DNase I through the bile duct (Materials and methods, Line 908-918). This method preserves the integrity of epitope, including CD3, CD4, CD8, CD19, CD45, and TCR δ , in immune cells derived from the pancreas (Supplementary Fig. 2C). As shown below, the epitope integrity of immune cells in the pancreas is dependent on the type of collagenase and concentration of trypsin inhibitor used (Response Fig. 3). However, the effect on CXCR3 integrity is unknown due to the lack of an appropriate positive control.

Response Fig. 3. Effect of collagenase type and the concentration of a trypsin inhibitor on epitope integrity in immune cells from the pancreas.

We thus did not show the results of the expression levels of CXCR3 in CD8 T cells in the pancreas. We could not show direct evidence for the migration of CXCR3⁺ CTLs to the pancreas from the PLN. However, a recent study using single-cell TCR-seq clearly showed that a subset of clones from stem-like CD8 T cells in the PLN gave rise to most of the pathogenic CD8 T cells in the pancreas (*Nature* 2022, <https://doi.org/10.1038/s41586-021-04248-x>).

- 11). It would be important to test cross priming in the absence of IGRP peptide as the assay will test direct antigen presentation in the presence of peptide, but not cross presentation. The authors should test NY8.3 T cells in response to NIT-1 cells in the presence of DCs but in absence of IGRP peptide. Alternatively, the authors need to pulse DCs with NIT-1 cells and use the pulsed DCs as antigen presenting cells to elicit NY8.3 T cell responses.
- 12). The authors used 10 $\mu\text{g/ml}$ IGRP peptide in their assays; this concentration of peptide is extremely high for NY8.3 CD8+ T cells.

Response to 11 and 12: We wish to express our deep appreciation of the Reviewer's insightful comment. First, we assessed the concentration of IGRP peptide in a culture of naïve 8.3 CD8 T cells. This revealed that a high concentration of IGRP peptide led to the spontaneous proliferation of 8.3 CD8 T cells without DCs (Response Fig. 4). On the basis of these results, we used 1 ng/mL of IGRP peptide for the co-culture experiments.

In our previous experiment, we used antigen-unloaded DCs for negative control, and therefore we could not examine the importance of the concentration of IGRP peptide for the culture of 8.3 CD8 T cells.

Response Fig. 4. Proliferation of naïve 8.3 CD8 T cells in the presence of IGRP peptide. MACS purified naïve 8.3 CD8 T cells were cultured with the indicated concentrations of IGRP peptide for 3 days.

Although our previous co-culture experiments using 10 $\mu\text{g/ml}$ of IGRP peptide revealed the impaired APC function of $Ty\text{k}2^{-/-}$ $\text{CD}8^+$ rDC, revised experiments using 1 ng/mL of IGRP peptide showed more clearly that $Ty\text{k}2^{-/-}$ $\text{CD}8^+$ rDC induced lower levels of 8.3 $\text{CD}8$ T cells expansion compared with $Ty\text{k}2^{+/+}$ $\text{CD}8^+$ rDC (Fig. 4F). This suggests that the impaired APC function in $Ty\text{k}2^{-/-}$ $\text{CD}8^+$ rDC was concealed by the high concentration of IGRP in our previous experiment.

Revised Fig. 4F. Proliferation of naïve 8.3 $\text{CD}8$ T cells in a culture of $\text{CD}8^+$ rDC and 1 ng/mL of IGRP peptide. (Left) Representative histogram and (right) proliferation of CTV labeled 8.3 $\text{CD}8$ T cells after co-culture with $\text{CD}8^+$ rDC and 1 ng/mL of IGRP peptide for 3d. P -values were calculated using two-tailed Student's t -test.

Next, we performed co-culture experiments using DCs, NIT-1 cells, and naïve 8.3 $\text{CD}8$ T cells without IGRP peptide. These cells were cultured for 3 days and the proliferation of 8.3 $\text{CD}8$ T cells was assessed. This experiment showed that $Ty\text{k}2^{-/-}$ $\text{CD}8^+$ rDC induced lower levels of 8.3 $\text{CD}8$ T cell expansion compared with $Ty\text{k}2^{+/+}$ $\text{CD}8^+$ rDC (Supplementary Fig. 5J). The proliferation efficiency (% divided) was reduced in this setting compared with a culture with IGRP peptide (Fig 4F). This may be due to the limited amount of IGRP epitope available from NIT-1 cells and/or to the reduced efficiency of antigen presentation by DC compared with conditions in the presence of IGRP peptide. Nevertheless, the impaired cross-priming of islet-autoreactive CTLs was evident in $Ty\text{k}2^{-/-}$ $\text{CD}8^+$ rDC compared with $Ty\text{k}2^{+/+}$ $\text{CD}8^+$ rDC.

Revised Supplementary Fig. 5J. Proliferation of naïve 8.3 CD8 T cells in a culture of purified CD8⁺ rDC and NIT-1 cells. (Left) Representative histogram and (right) proliferation of CTV labeled 8.3 CD8 T cells after co-culture with CD8⁺ rDC and NIT-1 cells without IGRP peptide for 3d. *P*-values were calculated using two-tailed Student's *t*-test.

On the basis of these results, we have revised text in the Results from:

“To assess the APC function of CD8⁺ rDC, we performed an *ex vivo* co-culture experiment. This revealed that *Tyk2*^{-/-} CD8⁺ rDC induced lower levels of 8.3 CD8 T cell expansion compared with *Tyk2*^{+/+} CD8⁺ rDC (Figure 4F).”

to

(Line 324-327) “To assess the APC function of CD8⁺ rDC, we performed an *ex vivo* co-culture experiment. This revealed that *Tyk2*^{-/-} CD8⁺ rDC (pulsed with IGRP peptide or NIT-1 cells) induced lower levels of 8.3 CD8 T cells expansion compared with *Tyk2*^{+/+} CD8⁺ rDC (Fig. 4F and Supplementary Fig. 5J).”

We performed an *ex vivo* cytotoxic assay in accordance with a previous study (Fig 5) (Cell & Bioscience 2014, <https://doi.org/10.1186/2045-3701-4-51>). In that study, the authors used 20 µg/mL of IGRP peptide for cytotoxic assays (8.3 CD8 T cells vs NIT-1 cells). In our study, we used 10 µg/mL of IGRP peptide for the assay, and showed the impaired cytotoxic function of *Tyk2*^{-/-} 8.3 CD8 T cells compared with *Tyk2*-sufficient 8.3 CD8 T cells (Fig. 5M).

Next, we have also reassessed the concentration of IGRP peptide in the cytotoxic assay. Similar results were obtained under conditions in which 1 ng/mL of IGRP peptide was used (Response Fig. 5, shown below). Thus, we confirmed that *Tyk2*^{-/-} CD8 T cells had an impaired cytotoxic function against β-cells compared with *Tyk2*-sufficient CD8 T cells irrespective of the IGRP peptide concentration.

Response Fig. 5. Killing assay with 1 ng/mL IGRP peptide (NIT-1:8.3 CTLs=1:1). Purified 8.3 CD8 T cells were stimulated with anti-CD3/CD28 beads and IL-2 in the presence or absence of IL-12 for 3d. *Ex vivo* activated 8.3 CD8 T cells and NIT-1 cells were co-cultured for 24 hours. The viability of NIT-1 cells was assessed using CellTiter 96 (Promega). *P*-values were calculated using two-tailed Student's *t*-test.

Our data suggest that the IFN- β signaling axis in CTLs has a role in the cytotoxic function of CTLs against β -cells (Fig. 5I, K). To exclude the possibility that treatment with IGRP peptide induced the expression of IFN- β in NIT-1 cells, we analyzed the gene expression levels of *Ifnb1* in NIT-1 cells after treatment with IGRP peptide. This showed that the expression of *Ifnb1* in NIT-1 cells was not induced by treatment with IGRP peptide (Response Fig. 6, shown below). Next, we analyzed the gene expression levels of *Ifnb1* in NIT-1 cells after cocultured with 8.3 CTLs. The expression levels of *Ifnb1* in NIT-1 cells were upregulated after coculture with 8.3 CTLs (Revised Fig. 5L, shown below). These results demonstrate that the source of IFN- β and the interaction between T cells and pancreatic β -cells elicits IFN- β from β -cells.

Response Fig. 6. *Ifnb1* expression levels in NIT-1 cells after treatment with IGRP peptide. *Ifnb1* expression in NIT-1 cells was measured 2 hours after treatment with the indicated concentrations of IGRP peptide. Data represent the with mean \pm SEM. *P*-values were calculated using one-way ANOVA with Dunnett's posttest.

Revised Fig. 5L. *Ifnb1* expression levels in NIT-1 cells after coculture with CTLs. NIT-1 cells were cultured with *ex vivo* activated 8.3 CD8 T cells and 10 $\mu\text{g}/\text{mL}$ of IGRP peptide. The expression of *Ifnb1* was determined by real-time qPCR. The relative quantification value is expressed as $2^{-\Delta\Delta C_t}$, where $\Delta\Delta C_t$ is the difference between the mean C_t value of duplicate measurements of the sample and the endogenous *Actb* control. P -values were calculated using one-way ANOVA with Dunnett's posttest.

On the basis of these results, we have added the following text in the Results.

(Line 406-408) "The expression levels of *Ifnb1* in NIT-1 cells were upregulated after coculturing with 8.3 CTLs (Fig. 5L)."

13). The authors showed that *Tyk2*^{-/-} IGRP CD8⁺ T cells had upregulated apoptosis-related genes including Casp3 and Birc5 (Fig.5I), it is not clear if those cells are indeed prone to apoptosis. Annexin V staining would provide some information and/or explanation as to why the cells upregulated apoptosis-related genes.

Response: In accordance with the Reviewer's comment, we performed annexin V staining in CD8 T cells derived from the pancreas (Supplementary Fig. 6F). Comparable annexin V binding to pancreatic CD8 T cells was observed between the *Tyk2* genotypes. This was discussed in the Results as shown below:

(Line 392-395) "However, binding of annexin V, a marker of early apoptosis, in IGRP CD8 T cells purified from the pancreas was comparable between the genotypes (Supplementary Fig. 5F), suggesting that *Tyk2*^{-/-} CD8 T cells were not conspicuously prone to apoptosis in the pancreas."

Further study is needed to reveal the role of TYK2 in the regulation of apoptosis in CD8 T cells.

F

Revised Supplementary Fig. 6F. Annexin V binding in IGRP-specific CD8 T cells in the pancreas.

P-values were calculated using two-tailed Student's *t*-test.

14). The earliest time when the wild type (or +/-) NOD mice developed T1D was around 14 to 15-wks-old (Fig.1A), whereas the earliest age when *Tyk2*^{-/-} NOD mice developed T1D was ~22-wk-old (Fig.1A). Early (from 6-wk-old) treatment of *Tyk2* inhibitor BMS-986165 in wild type NOD mice showed further delay in T1D onset, ~32-wk-old, and marked reduced T1D incidence, ~20% (Fig.6F), which is similar to the *Tyk2*^{-/-} mice shown in Fig.1A. *Tyk2* inhibitor treatment at pre-diabetic period (from 12 -wk-old) also strikingly delayed T1D onset (the first mouse developed diabetes was at 30-wk-old, Fig.6G). This reviewer assumes that if the group size were larger, especially the treatment group, which had only 4 mice, the statistical outcome could be significant, even though the overall incidence of diabetes was not reduced.

Response: We strongly appreciate the Reviewer's valuable comments on this point. We have described the timing of diabetes onset in mice and the limitation in the Results and Discussion as described below:

(Results; line 135-137) "In addition, the timing of diabetes onset was delayed in *Tyk2*^{-/-} mice (22 weeks of age (22w)) compared with their *Tyk2*^{+/+} (15w) and *Tyk2*^{+/-} (16w) littermates (Fig. 1A)."

(Results; line 458-459) "The timing of diabetes onset was delayed in the inhibitor treated mice (32w) compared with vehicle treated mice (19w) (Fig. 6F)."

(Results; line 461-463) "However, the timing of diabetes onset was delayed in the inhibitor treated mice (30w) compared with vehicle treated mice (20w) (Fig. 6G)."

(Discussion; line 522-529) "Even though the overall incidence of diabetes was not reduced, mice treated with the TYK2 inhibitor from 12w had a delayed onset of T1D compared with vehicle treated mice (Fig. 6G). These observations suggest that early intervention with a TYK2 inhibitor may be a potent immunotherapeutic strategy for autoimmune T1D, and treatment with the inhibitor in the late stage of

T1D may also be effective at suppressing the autoimmune process. The effect of the TYK2 inhibitor on autoimmune T1D will be better defined by studies using large group sizes.”

15). Many figure legends did not provide brief description about how the experiments were conducted. In this reviewer’s opinion, the figure legend should be more informative, and the readers should not need to go to and from the main text and/or Materials/Methods to find the information about the figures. Further, it is not clear why the authors used MFI as the readout for most, if not all, the flow cytometric analysis.

Response: In accordance with the Reviewer’s comment, we have revised the figure legends to be more informative.

The mean fluorescence intensity (MFI) is a numerical data reflecting the antigen expression levels. We used the MFI to compare the protein expression levels in cells between *Tyk2* genotypes. We think that the MFI is a widely used method for the analysis of protein expression levels in immunology research.

We wish to thank the Reviewer again for your valuable comments.

RESPONSES TO REVIEWER 2:

Dear Reviewer #2 (expert in JAK-STAT signaling and TYK2),

We wish to thank the Reviewer for your insightful comments on our paper. These have helped us to improve the quality of our paper. In particular, the revised figures related to the heatmap have improved the understandability of our data. Thank you very much for pointing this out.

The Reviewer's comments are written in blue. Our point-by-point responses to the Reviewer's comments are shown below.

Major comments:

- The authors should include in their introduction/discussion the relevant findings of Chandra et al (<https://doi.org/10.1038/s41467-022-34069-z>) in a human TYK2-deficient cell system

Response: We thank the Reviewer for this comment. We have cited the paper (Chandra *et al. Nat Commun.* 2022) in the Discussion. The reference number of this paper is 35 in our manuscript.

(line 505-509) “In contrast to these preservation effects of TYK2 inhibition in β -cells, a recent study suggested that TYK2 inhibition lead to reduced β -cell mass related to the defective formation of pancreatic endocrine precursors³⁵. Consequently, TYK2 may has a different role in β -cells depending on the stage of T1D and/or developmental stage of β -cells.”

(line 550-553) “In addition to the increased infection risk, there is a potential risk of lung cancer, non-Hodgkin lymphoma, and reduced β -cell mass with TYK2 inhibition^{35,67}, indicating that further safety assessments will be needed for long-term treatment with TYK2 inhibitors.”

We have also cited the paper in the Results for the rationale for the next experiment.

(line 212-214) “Type I IFN signaling in β -cells was associated with the presentation of autoantigens^{19,35}; therefore, we compared the expression levels of T1D-related autoantigens in β -cells.”

- Fig2C the authors should consider to complement the heat map by an alternative visualization of the crucial DEG and indicate in the results text more clearly which DEG (esp. IFNalpha and IFNbeta) might be of biological relevance albeit not reaching the applied statistical significance criteria. In this context the authors should also mention previous supportive work on human pancreatic beta-cells producing IFNalpha and CXCL10 under challenge conditions in a TYK2-dependent manner (Marroqui et al, <https://doi.org/10.2337/db15-0362>)

Response: We appreciate the reviewer's comment on this point. We have changed the heatmap to line graphs (Fig. 2D,E, shown below). The revised line graphs were created using the Z-score data used to create the previous heatmap.

Revised Fig. 2D, E. Expression levels of selected genes in β -cells.

We revised the Results so that our transcriptome data would be easier to understand.

(Line 190-201) “However, *Gbp2*, an ISG, was expressed at lower levels in β -cells from 11w *Tyk2*^{-/-} mice compared with 11w *Tyk2*^{+/+} mice (Fig. 2E). Notably, the expression level of *Fas* (*Cd95*), a cell surface death receptor involved in β -cell death³³, was lower in β -cells from 11w *Tyk2*^{-/-} mice compared with 11w *Tyk2*^{+/+} mice (Fig. 2E). The expressions of ISGs (*Gbp3*, *Bst2*, and *Isg20*), chemokines (*Cxcl9* and *Cxcl10*), a type I IFN receptor (*Ifnar2*), and type I IFNs (*Ifnb*, *Ifna1*) were slightly lower in β -cells from 11w *Tyk2*^{-/-} mice compared with 11w *Tyk2*^{+/+} mice, although they did not reach statistical significance (Fig. 2D, E). In contrast, *Cxcl13*, a ligand for CXCR5 involved in B cell chemotaxis, was expressed at higher levels in β -cells from 11w *Tyk2*^{-/-} mice compared with 11w *Tyk2*^{+/+} mice (Fig. 2D). These results suggest that type I IFN signaling, T cell migration into inflamed islets, and FAS-mediated β -cell death are impaired in β -cells of *Tyk2*^{-/-} mice compared with *Tyk2*-sufficient mice.”

In this context, we also revised the text and figures related to enrichment analysis (GO biological process: Fig. 2B, C). The revised figures clearly show the samples used for comparisons in each analysis.

(Line 182-189) “Gene ontology (GO) analysis revealed that β -cells from 11w *Tyk2*^{+/+} and *Tyk2*^{-/-} mice were enriched in immune-related pathways compared with those from 6w mice (Fig. 2B). Comparison of β -cells from 11w *Tyk2*^{+/+} and 11w *Tyk2*^{-/-} mice showed that both genotypes were enriched in immune-related pathways (Fig. C). Indeed, inflammation-related genes including *Vcam1*, *B2m*, and

interferon stimulated genes (*ISGs*), were highly expressed in β -cells from 11w mice compared with 6w mice irrespective of the *Tyk2* genotypes (Fig. 2D).”

Enriched in 11w WT (11w WT vs 6w WT)		Enriched in 11w WT (11w WT vs 11w KO)	
Pathways	p-value	Pathways	p-value
response to cytokine	5.82E-018	response to stress	8.50E-011
response to interferon gamma	2.26E-006	immune response	8.50E-006
leukocyte migration	9.44E-006	regulation of apoptotic process	1.63E-004
innate immune response	1.23E-005	response to cytokine	1.04E-002
antigen processing and presentation	6.78E-005		
response to interferon beta	2.04E-004	Enriched in 11w KO (11w WT vs 11w KO)	
cell type soecific apoptotic process	4.42E-003	Pathways	p-value
		immune response	5.72E-010
		response to stress	9.54E-009
		regulation of apoptotic process	2.02E-004
		inflammatory response	9.79E-003
Enriched in 11w KO (11w KO vs 6w KO)			
Pathways	p-value		
response to cytokine	2.43E-006		
regulation of immune response	6.89E-004		
innate immune response	1.54E-003		
positive regulation of leukocyte migration	1.64E-002		
response to interferon gamma	4.20E-002		
cellular response to interferon beta	4.20E-002		

Revised Fig. 2B, C. (B) Gene ontology (GO) biological process analysis of differentially expressed genes between β -cells from 11w mice and 6w mice (11w *Tyk2*^{+/+} vs 6w *Tyk2*^{+/+}, and 11w *Tyk2*^{-/-} vs 6w *Tyk2*^{-/-}).

Selected GO terms are shown. (C) GO biological process analysis of differentially expressed genes in β -cells from 11w mice (11w *Tyk2*^{+/+} vs 11w *Tyk2*^{-/-}). Selected GO terms are shown.

In accordance with the Reviewer’s comment, we have cited the paper (Marroqui *et al. Diabetes*. 2015) in the Results and Discussion. The reference number of this paper is 19 in our manuscript.

(Results, line 202-204) “The increased expression of MHC I in β -cells was reported to be a feature of autoimmune T1D, and TYK2 inhibition prevented IFN- α -induced MHC I expression in human β -cells^{19,34}.”

(Discussion, line 542-544) “In contrast, TYK2 inhibition reduces inflammation in β -cells in response to cytokines and a mimic of viral RNA, leading to the preservation of β -cells^{19,35}.”

• line 337ff, rationale for next experiments: the authors should acknowledge that impaired in vitro and in vivo killing capacity of TYK2-deficient CTLs has been already described (e.g. Simma *et al.*, <https://doi.org/10.1158/0008-5472.CAN-08-1705>).

Response: We have cited the paper (Simma *et al. Cancer Res.* 2009) in the Results regarding the rationale for the cytotoxicity assay. The reference number of this paper is 50 in our manuscript.

(Line 397-398) “A previous study reported that TYK2 had a role in CTL-mediated tumor surveillance⁵⁰.”

- line 477ff, discussion on potential risks/side effects of TYK2inib treatment: the authors should also mention the involvement of TYK2 in tumor surveillance and the potential cancer risk upon long-term deucravacitinib exposure (e.g. Yarmolinsky et al., <https://doi.org/10.1002/ijc.34180>).

Response: We thank the Reviewer for this comment. We have cited the paper (Yarmolinsky *et al. Cancer Epidemiology*. 2022) in the Discussion. The reference number of this paper is 67 in our manuscript.

(Line 550-553) “In addition to the increased infection risk, there is a potential risk of lung cancer, non-Hodgkin lymphoma, and reduced β -cell mass with TYK2 inhibition^{35,67}, indicating that further safety assessments will be needed for long-term treatment with TYK2 inhibitors.”

- Technical/methodological details necessary for the understanding of the presented data a scattered rather randomly in the manuscript; e.g. insulinitis scores 0-3 are defined in the legend to Figure S1 and not given in Material and Methods ,Histology‘; statistical stringency for RNASeq DEG is given in Material and Methods only and would help the reader if also mentioned in the results text or given in the legends of the main figures.

The authors are asked to revised the entire manuscript and indicate the necessary technical details in a clear and coherent manner.

Response: In accordance with the Reviewer’s comment, we have revised the entire manuscript including the figure legends. We feel that the comments have helped us improve our paper.

Minor comments:

Line 129-30: The journal strongly encourages researchers to follow the ‘Sex and Gender Equity in Research – SAGER – guidelines’ and to include sex and gender considerations for studies involving humans, vertebrate animals and cell lines where relevant to the topic of study (an overview can be found here). The authors should comment on their findings.

Response: In accordance with the Reviewer’s comment, we followed the SAGER guidelines. We have specified the sex of the NIT-1 cells and mice.

(Abstract, line 60-62) “Furthermore, we showed that treatment with BMS-986165, a selective TYK2 inhibitor, inhibited the expansion of CTLs, inflammation in β -cells, and the onset of autoimmune T1D in female NOD mice.”

(Method, line 834) “Female mice were used for all experiments.”

(Method, line 856-857) “NIT-1 cells, a pancreatic β -cell line derived from female NOD mice⁵¹, were obtained from ATCC (Manassas, VA, USA).”

In addition, we have added the following text to the Discussion.

(Line 529-530) “In addition, because our conclusions were based on experiments using female mice, further studies with male mice will be needed.”

Figure 4E: the number of labelled DEG confuses the take home message of the figure part, the authors could remove labels of non-ISGs and DEGs irrelevant of DC function

Response: We appreciate the Reviewer’s comment on this point. We mistakenly showed an incorrect volcano plot that did not fully represent the horizontal axis. We regret this oversight. The plot has been revised to show the full-length horizontal axis and genes of interest (Fig. 4E). ISGs: *Ifi44*, *Ifi103*, *Ifi2712a*, *Ifih1*, and *Irf9*. APC function associated genes: *Il4i1* and *Ar*.

We have revised the following text in the Results.

(Line 314-322) “We analyzed the gene expression profiles of CD8⁺ rDC in the PLN and found that *Interferon regulatory factor 9 (Irf9)* and ISGs including *Ifi2712a*, *Ifi44*, *Ifi203*, and *Ifih1* were highly expressed in *Tyk2*^{+/+} CD8⁺ rDC compared with *Tyk2*^{-/-} CD8⁺ rDC (Fig. 4E), suggesting that type I IFN signaling was upregulated in *Tyk2*^{+/+} CD8⁺ rDC (Supplementary Fig. 4K). *Tyk2*^{-/-} CD8⁺ rDC had higher expression levels of *Androgen receptor (Ar)*, which might be associated with the suppression of APC functions in DCs⁴⁵, and *Il4-induced 1 (Il4i1)*, which inhibits the proliferation of T cells⁴⁶, compared with *Tyk2*^{+/+} CD8⁺ rDC (Fig. 4E).”

Revised Fig. 4E. Volcano plot of transcriptome data of CD8⁺ rDC in the PLN of 6w mice (*Tyk2*^{+/+}, n=4;

Tyk2^{-/-}, n=4). CD8⁺ rDC were sorted and RNA-sequencing analysis was performed. Pairwise comparisons of differentially expressed genes were performed using DESeq2 and used to create the plot using ggVolcanoR (FDR < 0.1, logFC > |0.58|). ISGs and genes associated with APC function are highlighted.

We wish to thank the Reviewer again for your valuable comments.

REVIEWERS' COMMENTS

Reviewer #1 (expert in type-1 diabetes):

the authors have addressed the comments appropriately and this reviewer has no further questions.

Reviewer #2 (expert in JAK-STAT signalling and TYK2):

All reviewer's concerns and queries have been addressed appropriately and incorporated in the revised manuscript. The authors have performed a thorough work and the manuscript adds significant novelty to the field of autoimmune diseases connected to and driven by the JAK kinase TYK2.

Methodology and data analyses are appropriate and state-of-the-art, data presented and conclusions drawn are coherent well processed.

Acceptance recommended.

RESPONSE TO REVIEWER 1:

Dear Reviewer #1 (expert in type-1 diabetes),

The Reviewer #1's comment: The authors have addressed the comments appropriately and this reviewer has no further questions.

Response: We wish to thank the Reviewer for your insightful comments based on your expertise and giving us the opportunity to strengthen our manuscript. Again, thank you very much.

RESPONSE TO REVIEWER 2:

Dear Reviewer #2 (expert in JAK-STAT signaling and TYK2),

The Reviewer #2's comment: All reviewer's concerns and queries have been addressed appropriately and incorporated in the revised manuscript. The authors have performed a thorough work and the manuscript adds significant novelty to the field of autoimmune diseases connected to and driven by the JAK kinase TYK2. Methodology and data analyses are appropriate and state-of-the-art, data presented and conclusions drawn are coherent well processed. Acceptance recommended.

Response: We wish to thank the Reviewer for your valuable comments. We believe that these have greatly helped us to improve the quality of our paper. Again, thank you very much.